# Short-term acidification promotes diverse iron acquisition and conservation mechanisms in upwelling-associated phytoplankton

Robert H. Lampe [1,2], Tyler H. Coale[1,2], Kiefer O. Forsch[3], Loay J. Jabre [4], Samuel Kekuewa [3], Erin M. Bertrand [4], Aleš Horák [5,6], Miroslav Oborník[5,6], Ariel J. Rabines[1,2], Elden Rowland[4], Hong Zheng[2], Andreas J. Andersson[3], Katherine A. Barbeau [3] & Andrew E. Allen [1,2] ✉

Coastal upwelling regions are among the most productive marine ecosystems but may be threatened by amplified ocean acidification. Increased acidification is hypothesized to reduce iron bioavailability for phytoplankton thereby expanding iron limitation and impacting primary production. Here we show from community to molecular levels that phytoplankton in an upwelling region respond to short-term acidification exposure with iron uptake pathways and strategies that reduce cellular iron demand. A combined physiological and multi-omics approach was applied to trace metal clean incubations that introduced 1200 ppm $CO_2$ for up to four days. Although variable, molecular-level responses indicate a prioritization of iron uptake pathways that are less hindered by acidification and reductions in iron utilization. Growth, nutrient uptake, and community compositions remained largely unaffected suggesting that these mechanisms may confer short-term resistance to acidification; however, we speculate that cellular iron demand is only temporarily satisfied, and longer-term acidification exposure without increased iron inputs may result in increased iron stress.

Present-day atmospheric carbon dioxide ($CO_2$) concentrations of over 400 ppm have not been observed in the past 800,000 years and are increasing at an average rate that exceeds any observed in the past 20,000 years[1–3]. Approximately 30% of this released $CO_2$ is absorbed in the surface ocean resulting in a shift in the carbonic acid system and decline in seawater pH known as ocean acidification[4].

Coastal upwelling regions such as the California Current System (CCS) account for a significant proportion of marine primary production and exhibit some of the largest pH variability in the ocean due to the natural acidity of upwelled waters[5,6]. Some coastal upwelling

areas are also expected to experience accelerated or amplified ocean acidification[5,7]. In the CCS for example, the rate of acidification may be double that of the global average resulting in increased overall exposure to higher acidity for organisms, particularly during periods of strong upwelling[5,8,9].

Phytoplankton, especially diatoms, drive the high levels of primary production in upwelling regions that in turn support highly productive fisheries[5]; however, their vulnerability to acidification remains uncertain. Most phytoplankton species appear to exhibit relatively little response to acidification although there is variability

[1]Integrative Oceanography Division, Scripps Institution of Oceanography, University of California San Diego, 9500 Gilman Drive, La Jolla, CA 92093, USA. [2]Microbial and Environmental Genomics, J. Craig Venter Institute, 4120 Capricorn Lane, La Jolla, CA 92037, USA. [3]Geosciences Research Division, Scripps Institution of Oceanography, University of California San Diego, 9500 Gilman Drive, La Jolla, CA 92093, USA. [4]Department of Biology and Institute for Comparative Genomics, Dalhousie University, 1355 Oxford St, Halifax, NS B3H 4R2, Canada. [5]Biology Centre, Institute of Parasitology, Czech Academy of Sciences, 370 05 České Budějovice, CZ, Czechia. [6]Faculty of Science, University of South Bohemia, 370 05 České Budějovice, CZ, Czechia. ✉e-mail: aallen@jcvi.org

among taxa[10]. Field-based experiments with diatom communities suggest that effects from acidification are more frequently positive than negative and coastal communities are less affected than oceanic communities; however, shifts in community composition may also occur[11].

Further uncertainty in phytoplankton responses lies in the relationships between ocean acidification and iron bioavailability. Iron is a critical micronutrient for phytoplankton growth that is often limiting in the ocean including coastal upwelling regions such as the California Current and Peru/Humboldt Current Systems[12-15], and its availability in a future more acidic ocean remains controversial[16]. Reductions in bioavailable iron in response to acidification have been demonstrated in laboratory settings and are attributed to two mechanisms: (1) under acidic conditions, iron is less likely to disassociate from certain organic ligands to its more bioavailable form, labile dissolved inorganic iron (Fe′)[17], and (2) the algal inorganic iron uptake protein, phytotransferrin (pTF), is dependent on carbonate ion concentrations ($[CO_3^{2-}]$) that decline under acidic conditions[18]. In the California Current system, $[CO_3^{2-}]$ is estimated to have declined 35% during the 20th century[9].

The laboratory studies demonstrating these impacts in bioavailable iron may not have captured important dynamics including biotic interactions with siderophore-producing bacteria, a significantly more complex iron pool, and dynamic nutrient availability[19]. Further acidification experiments with natural seawater and a single diatom species, *Thalassiosira weissflogii*, did not produce significant differences in iron uptake rates indicating the presence of natural iron-binding ligands that are largely unaffected by pH[17], although reductions in Fe′ in the natural environment from increased complexation have also been observed[20,21]. Other experiments and modeling studies suggest that iron bioavailability may instead increase in response to acidification due to increased solubility of Fe(III)[22-24], increased oxidative lifetimes of highly bioavailable Fe(II)[5,25], and enhanced dissolution of particulate iron[26].

Previous field-based studies examining the effects of ocean acidification on phytoplankton communities under iron stress or limitation, namely within the iron-limited subarctic North Pacific and Southern Ocean, have shown conflicting results. Some experiments have shown that diatom communities are negatively impacted with reduced abundances, growth rates, and photosynthetic efficiencies[27-30]. Other experiments observed no changes in growth, macronutrient uptake, and community composition[31-33]. One experiment in the Southern Ocean observed no differences in growth or macronutrient uptake in the whole phytoplankton community; however, centric diatoms appeared to be favored over pennate diatoms indicating variable responses among more specific taxonomic groups that may not be evident if examining the entire community[34]. In contrast to these other experiments, Hopkinson, et al.[35] made pH adjustments to communities in the Gulf of Alaska resulting in very modest increases to growth and photosynthetic efficiency in conjunction with downregulation of certain photosynthetic proteins. Other ocean acidification studies have been conducted in regions with relatively high ambient iron concentrations, some of which used artificially high levels of strong iron-binding ligands to induce iron stress[36-39]; however, these ligands also increase the solubility of iron leading to increased overall dissolved iron concentrations[40].

Within upwelling regions, previous experiments also indicate a lack of change in response to acidification with diatom-dominated phytoplankton assemblages, but acidification was not assessed along with iron status even though the phytoplankton communities in these regions often exhibit high iron demand and limitation[12-14]. In the CCS, several experiments showed no discernible differences in growth, primary productivity, and carbon-to-nitrogen ratios[41-43]; however, responses were not evaluated with $CO_2$ levels greater than 800 ppm to exceed modern day surface pH variability in upwelling systems (Supplementary Tables 1 and 2). Only Osma, et al.[44] examined responses up

to 1,600 ppm $CO_2$ in the Peru/Humboldt Current System, but also did not observe differences in macronutrient drawdown, chlorophyll *a* concentrations, or diatom cell abundances.

High pH variability in coastal upwelling ecosystems has led to speculation that the native microorganisms in these regions may have the metabolic flexibility to accommodate relatively large pH changes. As evidenced by the response of higher trophic levels (mussels, gastropods, and planktonic copepods) in the Peru/Humboldt Current System, examining ocean acidification relative to local conditions rather than the global mean provides an improved understanding of organisms' tolerance[45]. As a result, phytoplankton responses in coastal upwelling regions may be different than those observed in other areas such as chronically iron-limited (high-nitrate low-chlorophyll) regions with low pH variability[5,46].

Furthermore, the aforementioned field-based experiments employed physiological assessments of the whole phytoplankton community that are incapable of discriminating changes only occurring within a specific taxonomic group. At times, taxon-specific pigment analyses and microscopic cell counts were employed showing a lack of change in abundances for more specific taxonomic groups or species, but not offering a window into taxon-specific physiological adjustments[28,31,35]. As previously mentioned, centric and pennate diatoms exhibited differing trends in their relative abundances in an experiment in the Southern Ocean[34] suggesting more taxonomically-specific resolution is needed to evaluate responses to ocean acidification. Transcriptomic and proteomic analyses can provide this greater resolution while also providing insight into the short-term molecular-level changes that may precede physiological changes or differences in community structure. Moreover, molecular-level knowledge of phytoplankton responses to acidification is limited to a small number of taxa under laboratory conditions without considering iron bioavailability or using organisms obtained from upwelling areas[47-49].

Thus, to evaluate potential impacts of acidification on phytoplankton in an upwelling region, ocean acidification was simulated with natural phytoplankton communities in the CCS. Trace metal clean techniques were used throughout enabling interrogation of acidification-driven effects on iron availability without unnatural changes to the iron pool. Air-$CO_2$ gas mixtures up to 1200 ppm $CO_2$ were applied, which exceeds both the projected global mean atmospheric $CO_2$ concentration for the year 2100 under the highest emission Shared Socio-Economic Pathway (SSP5-8.5) and most reported surface seawater $pCO_2$ and pH variability in the CCS (Supplementary Tables 1–3). With a combined physiological and multi-omic approach, these experiments enable probing of mixed phytoplankton assemblages' sensitivity to short-term acidification exposure from the whole community to taxonomically-specific molecular levels while considering the high importance of iron bioavailability.

## Results

### Experimental set-up, initial conditions, and assessments of iron limitation

Four incubation experiments were conducted in the CCS. Experiments 1–3 occurred within an upwelling filament, represented by newly upwelled waters at Experiment 1 that progressively aged by Experiment 3, and used communities from the near-surface (Fig. 1a and Supplementary Fig. 1, Table 1). Experiment 4 serves as a farther offshore site with an initial community from a subsurface chlorophyll maximum layer (SCML) at 60 m (Supplementary Fig. 1).

Initial pH at Experiment 1 was lowest (7.89 ± 0.00) as upwelled water naturally have low pH and then became less acidic (8.06 ± 0.01 by Experiments 3 and 4) from outgassing and primary production (Supplementary Fig. 2 and Table S3)[5,8,50,51]. In each experiment, triplicate bottles were supplemented with chelexed macronutrients (Table 1) and bubbled with HEPA-filtered air balanced with 400, 800, or 1200 ppm $CO_2$. Samples were then collected at one-day intervals

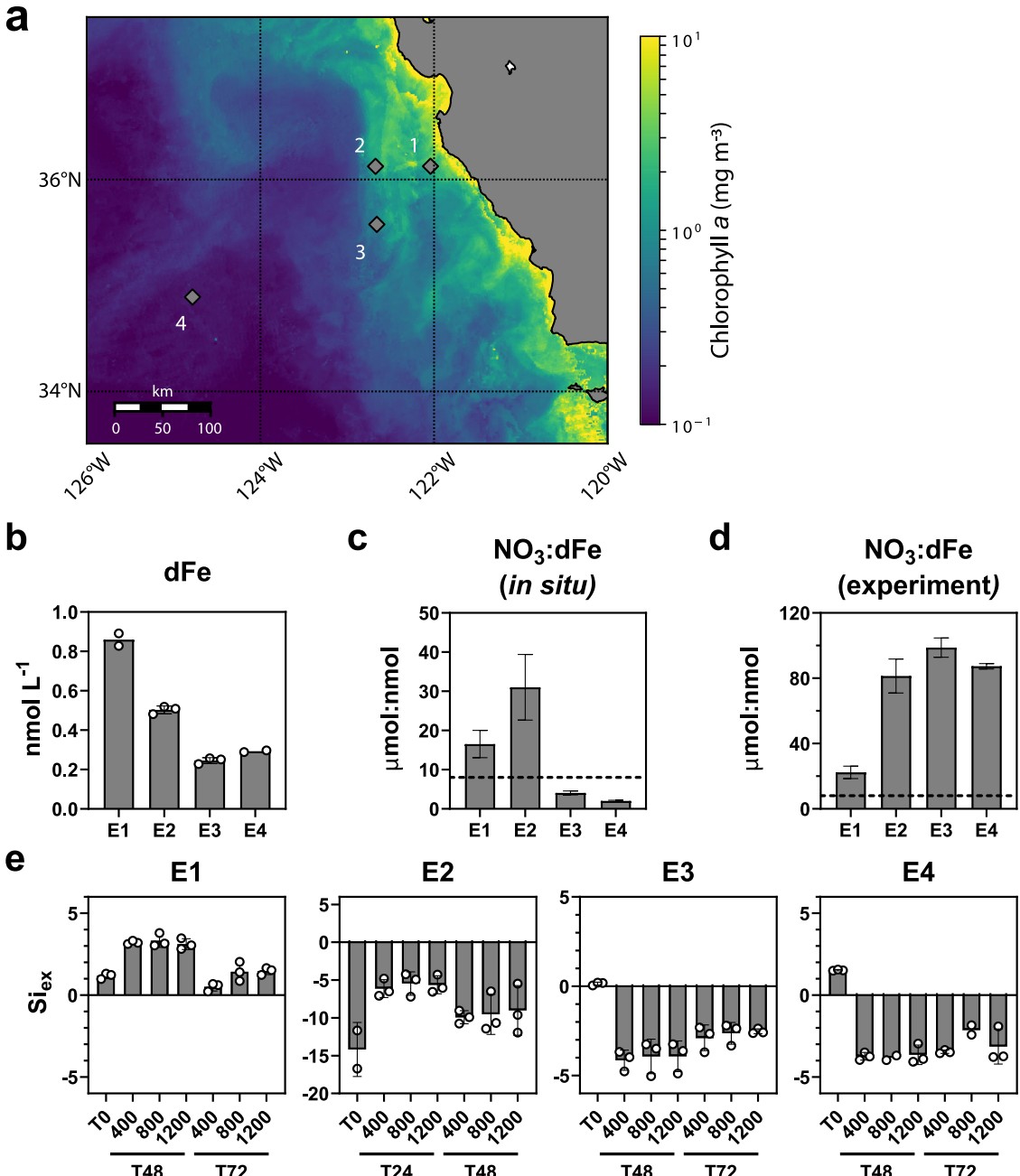

**Fig. 1 | Overview of the initial conditions for the four experiments (E1–E4).**
**a** Map of experiment locations with satellite-derived chlorophyll *a*. The chlorophyll *a* data was obtained from the NOAA Coastwatch Browser as a 14-day average centered on August 22, 2020 which falls between Experiments 2 and 3 (Table 1). **b** Dissolved iron concentrations (dFe, nmol $L^{-1}$). Nitrate ($NO_3$, μmol $L^{-1}$) to dissolved iron (dFe, nmol $L^{-1}$) ratios (**c**) before and (**d**) after macronutrient addition at the start of each experiment. The horizontal dashed line indicates the threshold for potential iron limitation (8 μmol:nmol)[53,122]. Ratios were derived and standard

deviations were propagated from the average values of field replicates presented in Figs. 1a and 2a. **e** $Si_{ex}$ proxy for diatom iron limitation within each experimental treatment. Initial time point (T0) values reflect the $Si_{ex}$ value following macronutrient addition. Negative values indicate iron stress or limitation due to preferential uptake of silicic acid ($H_4SiO_4$) relative to nitrate ($NO_3$)[53,122]. Error bars represent the standard deviation of the mean ($n = 3$ biologically independent samples or field replicate samples for T0). For dissolved iron concentrations at Experiments 1 and 4 and macronutrient data at T0 of Experiment 2, $n = 2$.

that varied among experiments to attempt to avoid macronutrient depletion with an upper limit of 96 h to also avoid increased bottle effects: 48 and 72 h for Experiment 1, 24 and 48 h for Experiment 2, and 48 and 96 h for Experiments 3 and 4 (Table 1). By the time of sampling, the incubations had an approximate pH of $8.0 \pm 0.0$, $7.8 \pm 0.0$, and $7.6 \pm 0.0$ (Supplementary Fig. 2) and $pCO_2$ concentrations of $413 \pm 21$, $800 \pm 34$, and $1197 \pm 44$ μatm for each respective treatment with some variability in $pCO_2$ concentrations among experiments (Supplementary Table 3).

As anticipated for the region, initial dissolved iron concentrations ([dFe]) were below 1 nmol $L^{-1}$ (Fig. 1b)[13]; only a small fraction of this dissolved iron exists as inorganic Fe(III) with most bound to organic ligands[52]. Labile particulate iron concentrations were highest in the vicinity of Experiment 1 ($>10$ nmol $L^{-1}$), and decreased to generally less than 1.0 nmol $L^{-1}$ offshore near Experiments 3 and 4 (Supplementary Fig. 3).

With these iron concentrations, the addition of macronutrients created the potential for iron stress or limitation, particularly at

**Table 1 | Experiment information**

| Exp. | Date | Latitude & Longitude | NO₃ & H₄SiO₄ added (µmol L⁻¹) | PO₄ added (µmol L⁻¹) | Time points (hours) |
|---|---|---|---|---|---|
| 1 | Aug. 13 | 36° 7.398 N 122° 2.475 W | 5 | 0.3125 | 48, 72 |
| 2 | Aug. 19 | 36° 7.333 N 122° 40.441 W | 20 | 1.25 | 24, 48 |
| 3 | Aug. 24 | 35° 34.823 N 122° 39.551 W | 25 | 1.5625 | 48, 96 |
| 4 | Aug. 30 | 34° 53.818 N 124° 46.905 W | 25 | 1.5625 | 48, 96 |

All experiments were conducted during the same cruise in August, 2019. Macronutrients were added at the concentrations specified. Subsampling was performed for all measurements at the time points listed (hours) in addition to the initial time point (T0). Chlorophyll $a$ concentrations were measured every 24 h.

Experiments 3 and 4 with Experiment 2 already showing signs of Fe limitation in situ. Ratios of nitrate to dissolved iron concentrations (NO₃:dFe) greater than eight (µmol:nmol) suggest the potential for iron limitation[53], and the initial conditions of all experiments exceeded this threshold following the addition of NO₃ (Fig. 1c, d). These ratios are also in line with those that naturally occur in the region[54].

Si$_{ex}$ is an additional biogeochemical proxy that serves as an indicator of Fe limitation in diatoms[53]; as Fe-limited diatoms preferentially uptake silicic acid (H₄SiO₄) relative to NO₃[55], lower than expected H₄SiO₄ concentrations compared to NO₃ concentrations indicates the presence of Fe limitation. Si$_{ex}$ reports the difference between H₄SiO₄ and NO₃ concentrations (*Methods*); therefore, negative values indicate greater H₄SiO₄ uptake due to iron stress in diatoms. At Experiment 2, negative values were observed from the initial timepoint (T0) throughout the experiment (Fig. 1e). Following macronutrient addition, negative values were also observed in Experiments 3 and 4 while values remained positive in Experiment 1 (Fig. 1e).

Parallel incubations with unamended control treatments and iron addition treatments (+5 nmol L⁻¹) were performed to further evaluate the presence of iron stress during the experiments. Following 24 h of incubation at Experiment 2 and 48 h at Experiment 3, chlorophyll $a$ concentrations significantly increased with the addition of iron indicating the presence of iron limitation in these communities by the first sampling time points (Supplementary Fig. 4). At Experiment 4, nitrate was initially depleted indicating no iron limitation in the initial community (Fig. 2 and Supplementary Fig. 5); however, in incubations initiated 24 h later while following a Lagrangian drifter, significant increases in chlorophyll $a$ were observed only with the addition of both iron and nitrate suggesting co-limitation in this community. As nitrate was added without iron in the community subjected to increased acidification, it was likely iron-limited. Considering low (<1 µmol L⁻¹) nitrate concentrations after 72 h in Experiment 1 (Fig. 2), the phytoplankton community there was not iron-limited, although it may have encountered some degree of iron stress.

**Whole-community physiological responses**
Despite differences in pH between treatments at all timepoints (Supplementary Fig. 2), few significant differences (P < 0.05) were observed in chlorophyll $a$ concentrations, particulate carbon-to-nitrogen ratios, and macronutrient concentrations. Short-term iron uptake rates were normalized to both chlorophyll $a$ and particulate organic carbon (POC) with both inorganic iron (FeCl₃) and iron bound to the organic ligand desferrioxamine B (Fe-DFB)(Fig. 2 and Supplementary Figs. 6, 7). Although chlorophyll content varies in response to environmental conditions[56], light and temperature were consistent during the experiments resulting in comparable chlorophyll-normalized rates that may better reflect photoautotrophic biomass compared to POC. Chlorophyll $a$ has often also been used to normalize Fe uptake rates in phytoplankton communities as POC includes heterotrophic biomass[39,57,58].

When normalized to chlorophyll $a$, maximum inorganic Fe uptake rates were significantly higher in the 1200 ppm when compared to the 400 ppm treatment at the final time point for Experiment 3 (P = 0.016) indicating an increase in cell surface transporters[57]. All other Fe uptake rates were not significantly different among treatments at each time point and experiment including when normalized to POC. The measurable uptake of Fe-DFB, particularly at Experiments 3 and 4 where rates were similar to those of iron-limited populations and isolates[59,60], further indicates iron stress as phytoplankton normally only acquire significant amounts of Fe-DFB when iron-limited[57].

Community growth rates (µ) estimated from chlorophyll $a$ concentrations were also not significantly different between the experiments (Supplementary Fig. 8). In Experiment 1, a relatively rapid average growth rate of 0.87 ± 0.10 d⁻¹ suggests that members of these communities may have performed multiple cell divisions leading to a large increase in chlorophyll $a$ and drawdown of NO₃ to <1 µmol L⁻¹. Multiple divisions also may have potentially occurred at Experiment 3 (µ = 0.46 ± 0.04 d⁻¹) and approximately one division may have occurred before the termination of Experiment 4 (µ = 0.27 ± 0.04 d⁻¹) as both experiments were four days long. Collectively, these rates suggest that the total exposure to acidification exceeds that of the community doubling time. In the case of Experiment 2, the shorter duration of the experiment (48 h) and lower average growth rate (µ = 0.32 ± 0.08 d⁻¹) suggests that the community doubling time exceeded the duration of exposure. This lower growth rate was likely driven by iron-limitation in the initial community

**Microbial community composition**
Eukaryotic and prokaryotic community compositions throughout the ocean acidification experiments were inferred via 18S and 16S ribosomal RNA (rRNA) amplicon sequencing (Fig. 3, Supplementary Datasets 1 and 2). Within eukaryotic phytoplankton rRNA, Experiments 1–3 were dominated by centric diatoms, particularly the large chain-forming genera *Chaetoceros, Eucampia*, and *Thalassiosira*, with relatively high abundances in Experiment 4 as well (Fig. 3a, Supplementary Fig. 9). Pennate diatoms were also present throughout and largely consisted of *Pseudo-nitzschia* with a notable presence of *Navicula* in Experiment 2 (Supplementary Fig. 9). *Chaetoceros, Thalassiosira*, and *Pseudo-nitzschia* are also among the most abundant diatoms in both the region and the ocean[61,62]. Experiment 3 had a high abundance of the prasinophyte *Ostreococcus* (Bathycoccaceae) and the pelagophyte *Pelagomonas calceolata* (Pelagomonadales), both of which are cosmopolitan picoeukaryotes[63,64]. *Pelagomonas calceolata* further dominated Experiment 4 in line with their previously observed high abundances in SCMLs[65].

Alphaproteobacteria and Gammaproteobacteria, were prevalent in all four experiments as observed in the global ocean (Fig. 3a)[66]. In particular, these groups were dominated by the ubiquitous *Rhodobacterales* order[67] and SAR86 clade[68] (Supplementary Fig. 10). Cyanobacteria were less than 0.5% of the prokaryotic communities in Experiments 1 and 2 further indicating that the phytoplankton communities were dominated by diatoms. Experiment 3 had a notable presence of *Synechococcus* (14% of 16S rRNA), and Experiment 4 was dominated by *Prochloroccocus* (71% of 16S rRNA) indicating a higher presence of cyanobacteria in both of those experiments.

Like the physiological measurements, increased CO₂ largely did not alter the eukaryotic or prokaryotic community compositions. Alpha diversity expressed as the Shannon Index was not significantly different among CO₂ treatments in each experiment except for prokaryotes in Experiment 1 where diversity was higher in the 400 ppm treatment (Fig. 3b). Community dissimilarity (Bray-Curtis) was driven more by the distinct initial communities and change in the communities over time with no significant differences among CO₂ treatments

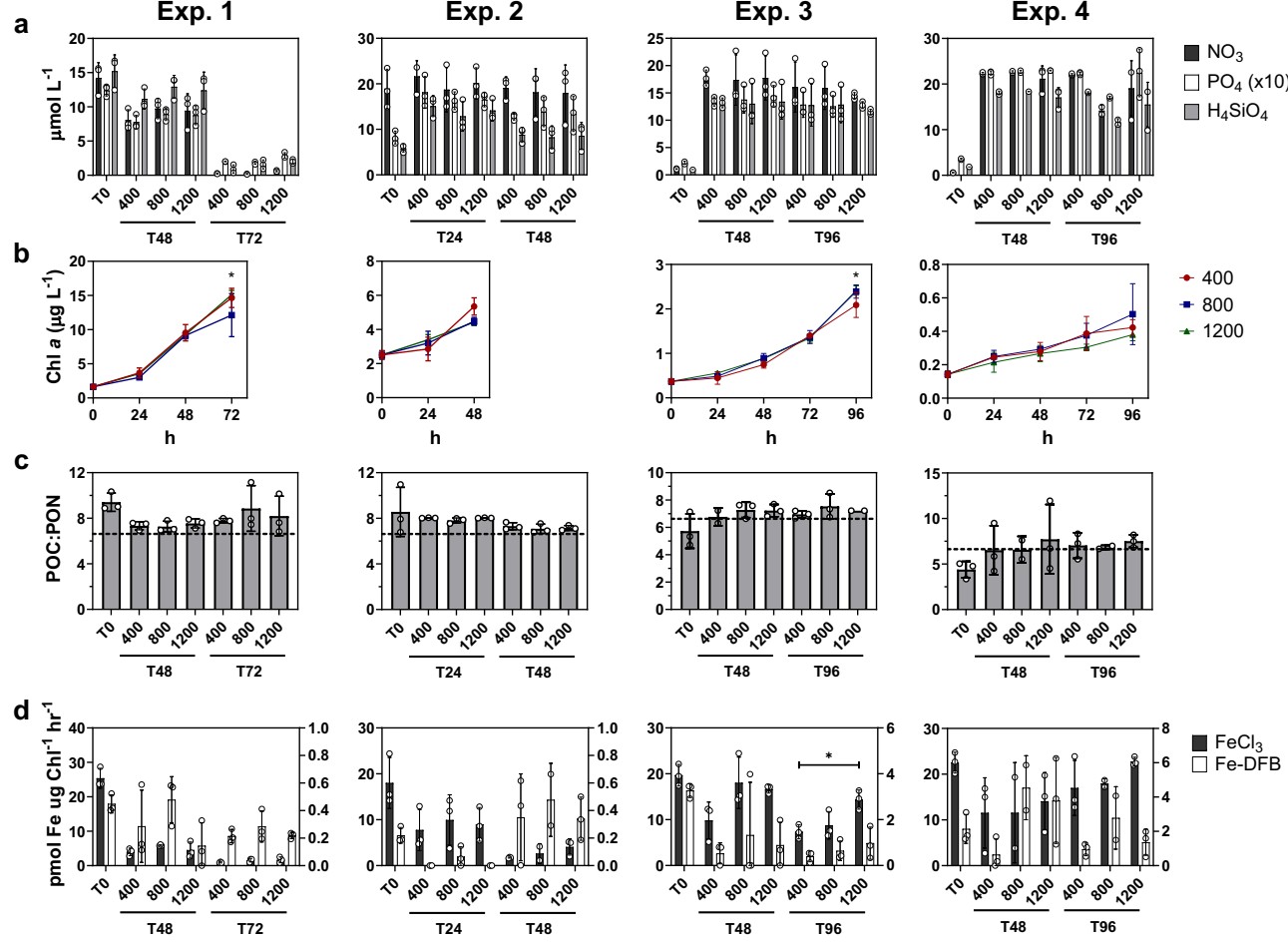

**Fig. 2 | Physiological measurements associated with the incubations. a** Nitrate (NO₃, black), phosphate (PO₄ x 10, white), and silicate (H₄SiO₄, gray) concentrations (μmol L⁻¹). The initial (T0) concentrations are those before amending at the concentrations shown in Table 1. **b** Chlorophyll *a* concentrations (μmol L⁻¹) from every 24 h interval for the 400 (red), 800 (blue), and 1200 ppm CO₂ treatments (green). At Experiment 1, the 800 ppm and 1200 ppm treatments are significantly different (*P* = 0.026) after 72 h. At Experiment 3, the 400 ppm treatment is significantly different from both the 800 ppm (*P* = 0.018) and 1200 ppm treatments (*P* = 0.010) after 96 h. **c** Ratios of particulate organic carbon (POC) to particulate organic nitrogen (PON). The expected Redfield ratio (106:16) is denoted with a

horizontal dashed line. **d** Short-term iron-59 uptake rates normalized to chlorophyll *a* (μmol Fe mol C⁻¹ hr⁻¹) (pmol Fe μg Chl⁻¹ hr⁻¹) for inorganic iron (FeCl₃, black, left y-axis) and organically-complexed iron as ferrioxamine B (Fe-DFB, white, right y-axis). At Experiment 3, the FeCl₃ uptake rates are significantly different between the 400 and 1200 ppm treatments (*P* = 0.016). Both pCO₂ treatments and time points in hours are denoted on the x-axis in (**a, c, d**). Error bars represent the standard deviation of the mean (*n* = 3 biologically independent samples or field replicate samples for T0 except the Experiment 4800 ppm samples where *n* = 2). Significant differences (*P* < 0.05, Two-way ANOVA with Tukey's HSD test) are denoted with an asterisk (*).

---

for both prokaryotes and eukaryotes in each experiment (Fig. 3c and Supplementary Fig. 11, PERMANOVA, *P* > 0.05).

Differential abundances of specific taxa as rRNA-derived amplicon sequence variants (ASVs) were evaluated with DESeq2 among the treatments at the final timepoints (Fig. 3d and Supplementary Fig. 12). Between the 400 ppm and 1200 ppm treatments, DESeq2 estimated that fewer than 1.5% of the eukaryotic ASVs and no more than a single prokaryotic ASV within each experiment were differentially abundant. Five 18S ASVs with significantly decreased abundances at 1200 ppm were haptophytes belonging to the Prymnesiophyceae class or Isochrysidales order without more detailed taxonomic resolution. These taxonomic groups contain calcifying algae that may be more susceptible to acidification as the precipitation of calcium carbonate becomes less energetically favorable, although a wide range of responses for haptophytes has previously been observed and some species are not calcifying[10,69,70]. Three additional differentially abundant 18S ASVs including a hydrozoan and a copepod, which likely stems from uneven distributions of metazoans between treatments rather than an acidification-driven response.

## Differential expression of transcripts and proteins among CO₂ treatments

Dominant eukaryotic phytoplankton groups identified from 18S rRNA were abundant in corresponding poly(A)-selected metatranscriptomes and metaproteomes (Fig. 3a). Between 23.8% and 48.2% of orthologous gene groups, or orthogroups, were unique to the detected protein sequences in each experiment's metatranscriptome assembly highlighting the diverse functional repertoires within each community (Supplementary Fig. 13) in addition to their significant differences in microbial community composition (Fig. 3c). In each experiment, differential expression of KEGG-annotated transcripts was examined between CO₂ treatments within these taxonomic groups. The same was conducted for protein abundances in Experiments 1 and 3. Often differentially expressed transcripts were not detected as proteins (Supplementary Table 4), likely due to comparatively lower resolution with mass spectrometry, but some consistent patterns were observed.

Transcriptional changes were generally lower at 800 versus 400 ppm compared to 1200 versus 400 ppm indicating that transcriptional responses are dependent on CO₂ conditions (Fig. 4a and

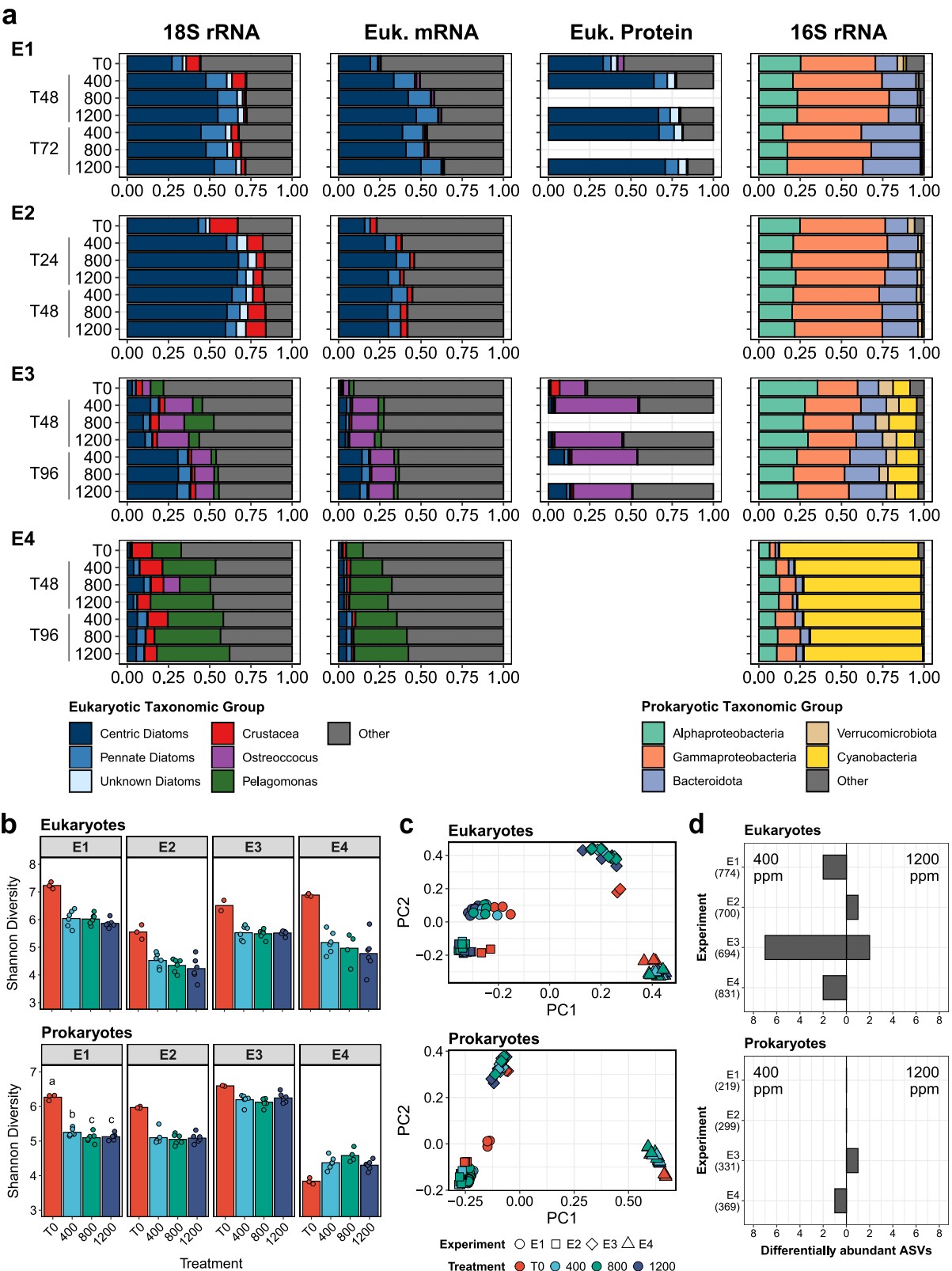

Supplementary Table 5); protein abundances were not examined for the 800 ppm treatments. Dominant taxonomic groups also displayed differing levels of responsiveness with centric diatoms being more responsive than pennate diatoms in terms of both transcripts and proteins. Further variability was observed in the transcripts or proteins that were differentially abundant within taxonomic groups across experiments and time points.

As acidification is hypothesized to induce iron stress, three indicators of iron stress, the iron starvation-induced genes *ISIP1*, *ISIP2A/pTF*, and *ISIP3*, were examined in both centric and pennate diatoms (Fig. 4b and Supplementary Fig. 14). ISIP1 is linked to endocytosis of siderophore-bound iron[71]. ISIP2A, or phytotransferrin (pTF), is an inorganic iron uptake protein with a direct dependence on carbonate ions[18]. As previously mentioned, this carbonate ion dependence may

**Fig. 3 | Microbial community composition and diversity for each experiment (E1-4) from both 18S and 16S rRNA amplicons. a** The average relative abundances of major taxonomic groups from 18S rRNA, poly(A)-selected mRNA (metatranscriptome), proteins (metaproteome), and 16S rRNA within each time point and treatment. The average abundances from the metatranscriptomes were derived from the relative abundances of eukaryotic-assigned transcripts per million normalized reads (TPM). The average abundances from the metaproteomes were derived from the average relative abundances of total sum ion intensities for all eukaryotic peptides identified. **b** Alpha diversity expressed as the Shannon Index within each sample for eukaryotic rRNA (top) and prokaryotic rRNA (bottom). Bars indicate the average Shannon Index. For prokaryotic ASVs at Experiment 1, the initial conditions (T0) are significantly different than all other treatments (Kruskal-Wallis test, Benjamini & Hochberg adjusted $P = 0.037$). The 400 ppm treatment is also significantly different from both the 800 ppm (Benjamini & Hochberg adjusted $P = 0.037$) and 1200 ppm (Benjamini & Hochberg adjusted $P = 0.045$) treatments. $n = 3$ biologically independent samples or field replicate samples for T0 except the Experiment 3 T0 and Experiment 4 800 ppm samples where $n = 2$. **c** PCoA plots of Bray-Curtis dissimilarity for eukaryotic rRNA (top) and prokaryotic rRNA (bottom). Experiments and treatments are denoted by color and shape respectively as described in the legend. **d** The number of significantly differentially abundant amplicon sequence variants (ASVs) at each experiment determined by DESeq2 (Wald test) between the 400 ppm (left) and 1200 ppm (right) treatments at the final time point for each experiment (Benjamini & Hochberg adjusted $P < 0.05$). Eukaryotic ASVs are shown on top and prokaryotic ASVs are shown on the bottom plot. The total number of ASVs for each experiment is shown in parentheses next to the experiment number of the y-axis.

result in negative impacts on iron uptake as carbonate ions decline under acidic conditions. ISIP3 does not have a known function although it has a ferritin-like domain further suggesting a role in iron metabolism[72].

These genes are often highly responsive to iron status in diatoms[72–74], and high gene expression relative to others in each sample would not be expected unless the cells were experiencing iron stress or limitation. In Experiments 1 and 2, all three genes were in the top 1% of expressed transcripts by the final time point in all $CO_2$ treatments indicating extremely high expression, iron stress, and iron demand even though macronutrients were depleted at the end of Experiment 1 (Fig. 4b and Supplementary Fig. 14). In Experiments 3 and 4, transcript expression was within the top 5% of expressed genes also indicating iron stress. Importantly, none of the *ISIP* transcripts were significantly differentially expressed between $CO_2$ treatments at each time point ($P > 0.05$) suggesting no relative increase or decrease in iron stress as a function of $CO_2$. These patterns were also consistent between centric and pennate diatoms (Fig. 4b and Supplementary Fig. 14).

ISIP proteins were not universally detected; however, in Experiments 1 and 3, ISIP3 from centric diatoms was in the top 5% of detected proteins at the final time points further suggesting the development of iron stress in these experiments (Supplementary Fig. 15). Other ISIP proteins approached the top 5% in both centric and pennate diatoms. Protein abundances were also not significantly different between $CO_2$ treatments. The exceptions are for ISIP1 and ISIP2A in centric diatoms in Experiment 3 at 48 h where they were not detected in the 1200 ppm treatment; however, both were detected after 96 h where abundances were not significantly different (Supplementary Fig. 15).

Nutrient stress in Experiments 1 and 3 was further evaluated by examining cellular resource allocations to ribosomal and photosynthetic proteins (Supplementary Fig. 16)[75]. At Experiment 1, both centric and pennate diatoms displayed significant reductions in their ribosomal protein mass fractions from 48 to 72 h, likely reflecting the onset of nitrogen limitation after 72 h ($P < 0.01$, Supplementary Fig. 16). At the earlier timepoint, 48 h, pennate diatoms displayed moderate increases in their ribosomal protein mass fractions under higher $CO_2$ ($P = 0.11$) indicating an increase in ribosomal investment from increased acidity. At Experiment 3, a different response was observed in centric diatoms and *Ostreococcus* with both significantly reducing their photosynthetic protein mass fractions after the initial time point ($P < 0.01$, Supplementary Fig. 16). This shift is likely a result of increased iron stress as observed in diatoms within the Southern Ocean[75]. A lack of change to the ribosomal mass fraction in comparison to pennate diatoms at Experiment 1 further highlights the potential differences in responses among taxonomic groups or the interactive effects of ocean acidification under increased Fe stress.

The total number of significantly differentially expressed transcripts was variable across the experiments with the most found at Experiments 1 and 3 where the exposure to acidification relative to the community growth rates were the longest (Supplementary Fig. 8,

Supplementary Tables 4 and 5). Although total differential expression of transcripts within each taxonomic group was less than 1%, those that were differentially expressed highlight the reconfiguration of iron, nitrogen, and carbon uptake proteins as well as cellular iron requirements to increased $CO_2$. Many other differentially expressed genes are of unknown function as anticipated for eukaryotic phytoplankton[76].

Under higher $CO_2$, increased expression of genes related to iron uptake or transport that are distinct from *ISIP2a/pTF* was observed (Fig. 4c). At times, several of these genes have been observed to be upregulated by diatoms under iron stress in the laboratory and field[73,77,78]. This response includes an iron permease (*FTR1*). Diatom FTR1 proteins are related to those found in green algae and yeast where they have been shown to mediate inorganic ferric iron uptake like *pTF* (Supplementary Fig. 17)[72,79]. *FTR1* transcripts and proteins significantly increased in centric diatoms at 1200 ppm $CO_2$ in Experiment 3 indicating an increase in cell surface transporters. This increase in transporters likely resulted in the significantly higher maximum uptake rates of inorganic iron observed here (Fig. 2d)[80].

Two putative ferrous iron transporters displayed increased transcript expression in Experiment 1 under increased $CO_2$: one in the ZIP family (ZIP1) in centric diatoms and one in the CDF family (FieF-like) in pennate diatoms (Fig. 4c). Diatom homologs of the *ZIP1* gene are related to ZIP-family genes in green algae, and have been found to be upregulated under Fe stress for diatoms in the both the field and laboratory[73,77,81]. The CDF transporter found here is from a distinct clade than one previously described to be upregulated under Fe stress in the model diatom *P. tricornutum* (Supplementary Fig. 18) and is related to green algal CDF transporters that have been shown to have Mn transport capability[82].

ZIP and CDF-family transporters can transport multiple divalent metal cations[79]; therefore, if localized to the cell surface, they may serve as low affinity Fe(II) permeases to take advantage of potentially increased Fe(II)[25,83]. Direct Fe(II) uptake has been demonstrated in eukaryotic microalgae including diatoms supporting this role[84]. ZIP and CDF-family transporters may also be involved with intracellular transport or release of stored iron potentially indicating an increased reliance on iron recycling or storage[79,81,85].

Putative siderophore-bound iron uptake genes, an FecCD-domain containing gene in centric diatoms and SLC49 family transporters in pennate diatoms also displayed increased transcript expression at 1200 ppm $CO_2$ in Experiment 1 (Fig. 4c). These genes are poorly characterized in diatoms but have been linked to siderophore-based Fe uptake in other organisms. The FecCD domain has been characterized as part of a bacterial operon for a ferric citrate ABC transporter in *E. coli*[86]. Similar proteins are present in several centric and pennate diatoms and were likely obtained via horizontal gene transfer from bacteria (Supplementary Fig. 19). Diatom SLC49-family transporters are similar to those found in metazoans where they have been functionally characterized (Supplementary Fig. 20)[87]. These proteins are related to heme transport and may be responsible for heme uptake

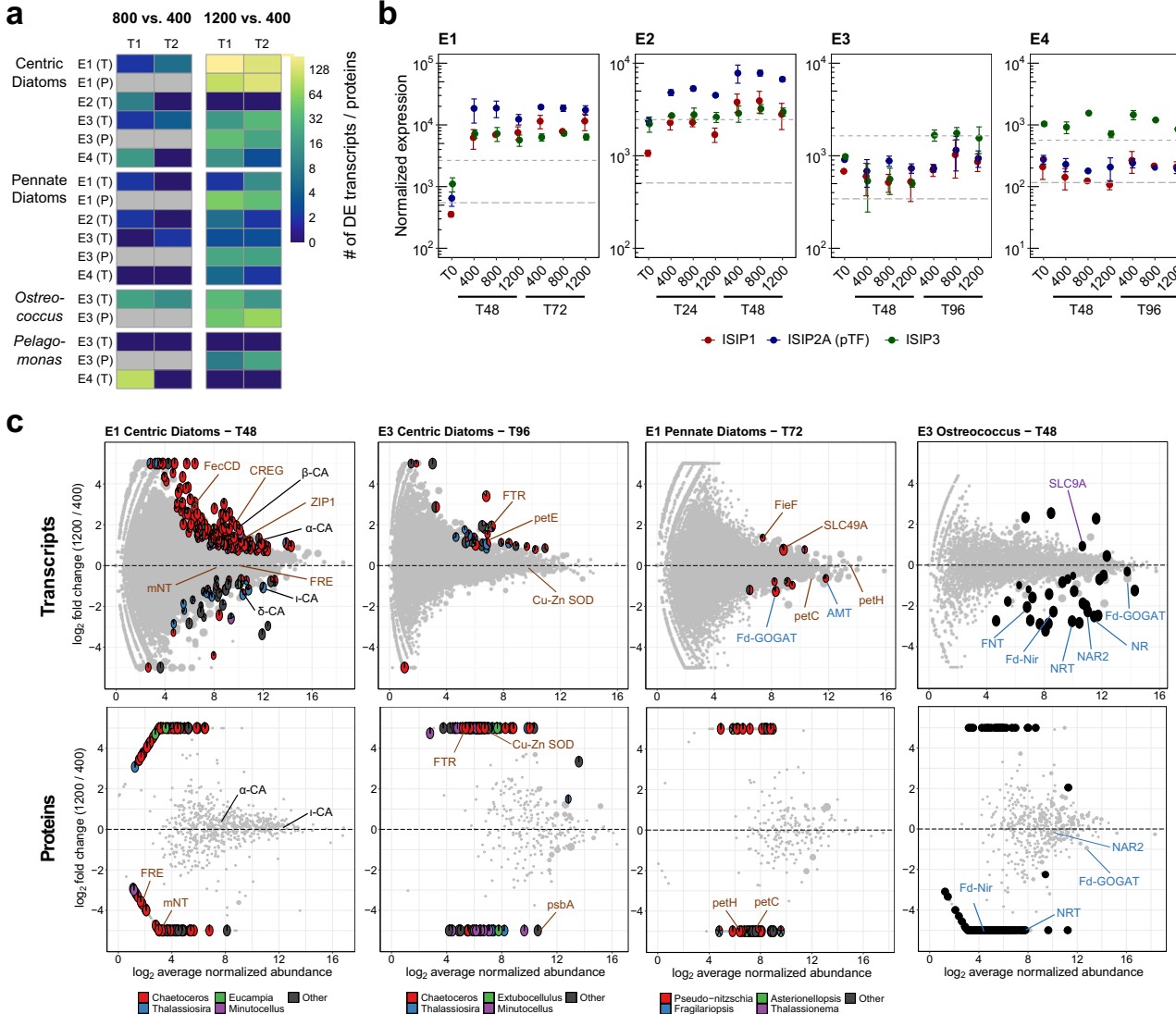

**Fig. 4 | Molecular-level responses to increased pCO₂. a** Heatmap of the number of significantly differentially expressed transcripts (T) or proteins (P) for dominant taxonomic groups at each experiment (DESeq2 Wald test, $P < 0.05$, log₂ scale) in the 800 ppm vs 400 ppm treatments (left) and 1200 ppm vs 400 ppm treatments (right) at each time point. Gray boxes indicate time points where there is no corresponding proteome. **b** Normalized transcript expression of iron starvation-induced proteins (ISIPs) in centric diatoms (log₁₀ scale): *ISIP1* (red), *ISIP2A* (phytotransferrin or *pTF*, blue), *ISIP3* (green). The horizontal dashed lines indicate the averaged 95th and 99th percentile for total transcript abundance in each experiment. Error bars represent the standard deviation of the mean ($n = 3$ biologically independent samples or field replicate samples for T0 except the Experiment 3 T0 and Experiment 4 800 ppm samples where $n = 2$). **c** MA plots for taxonomic groups, experiments (E), and time points (T), where significant differential expression for transcripts (top row) or proteins (bottom row) of interest between the 1200 and 400 ppm treatments was observed. Differentially expressed transcripts or proteins (DESeq2 Wald test, Benjamini & Hochberg adjusted $P < 0.05$) are shown as pies representing the proportion of expression for dominant genera. Pie size is proportional to $P$ value with larger pies denoting a smaller $P$ value. Black circles rather than pies are shown for *Ostreococcus* as expression is already examined at the genus level. Gray points are genes that were not significantly differentially abundant (DESeq2 Wald test, Benjamini & Hochberg adjusted $P > 0.05$). Data points with log₂ fold changes less than −5 or greater than 5 are plotted at −5 and 5 respectively.

as observed in diatom cultures and eukaryotic phytoplankton in the natural environment[87,88].

The increased expression of these transporters is consistent with the expected decrease in bioavailable free inorganic ferric iron that may result from acidification. At Experiment 1, increased transporter expression implies an increased reliance on siderophore-bound iron or ferrous iron that may increase unlike ferric iron. At Experiment 3, the response was instead an increase of a ferric iron permease (FTR1) that does not have a carbonate-dependence like phytotransferrin. Observed as both transcripts and proteins, this increase suggests that there were a greater number of cell surface transporters for inorganic ferric iron as supported by higher maximum rates of inorganic iron uptake (Fig. 2d).

One exception to this increase in transporter expression was in centric diatoms at Experiment 1 where decreased abundances of ferric reductase proteins under increased CO₂ were observed (Fig. 4c). Ferric reductases have been implicated as part of a siderophore uptake pathway including that of Fe-DFB in diatoms[89], suggesting that this other pathway may not be favored under these conditions; however, the ones expressed here may represent separate ferric reductase homologs that do not serve the same functional role. Ferrichrome-binding protein 1 (FBP1) acts as a receptor in concert with ferric reductases for siderophore uptake including Fe-DFB, but it was not differentially expressed (Supplementary Fig. 21 and 22) corresponding to the similar Fe-DFB uptake rates among treatments (Fig. 2d and Supplementary Fig. 6).

As some of the ferrous iron transporters with increased expression may be responsible for intracellular transport, these increases may also suggest a greater reliance on iron recycling or releasing stored iron to support growth in the short-term. Also in response to increased $CO_2$ at Experiment 1, centric diatoms increased transcript expression of a CREG-like gene (cellular repressor of E1-A stimulated genes; Fig. 4c) that is upregulated under iron stress in diatoms[77,78,90]. Diatom CREG proteins are proposed to have a regulatory role in positively inducing endocytosis of iron when iron availability is low suggesting an increased dependence on endocytosis of iron under increased $CO_2$[91].

Other differentially expressed genes indicate an adoption of strategies that reduce cellular iron demand. Due to its versatile redox chemistry, iron plays an essential role in photosynthesis-related proteins in diatoms, and reductions in photosynthetic proteins can be related to iron stress[83]. In centric diatoms, higher $CO_2$ reduced protein abundances of the photosynthetic proteins cytochrome b6 (petB) at Experiment 1 as well as apocytochrome f (petA) and photosystem II D1 (psbA) at Experiment 3 (Fig. 4c and Supplementary Fig. 23). Similarly, these proteins had lower abundances due to increased acidification in two previously mentioned low iron experiments in the Gulf of Alaska[35]. In pennate diatoms, lower protein abundances for the cytochrome $b_6f$ complex protein, petC, and the ferredoxin-NADP+ reductase, petH, were also found at Experiment 1.

Iron is also critical for mitochondrial electron transport[83], and in centric diatoms at Experiment 1, downregulation of an iron-containing mitochondrial protein, mitoNEET (mNT), that may be involved in signaling was observed (Fig. 4c)[92]. MitoNEET sequences were found to be widely distributed in diatoms where they are related to those in other eukaryotic phytoplankton including green algae (Supplementary Fig. 24). In the model green algae, *Chlamydomonas reinhardtii*, mNT was downregulated under iron stress indicating that this iron-sparing response may be more widely distributed among phytoplankton[93].

Cellular iron requirements can also be reduced by using iron-free functionally equivalent proteins. In Experiment 3, centric diatoms increased transcript abundances of plastocyanin (petE), a copper-containing functional equivalent for the iron-requiring photosynthetic electron transport chain protein cytochrome $c_6$[94], and increased protein abundances of the Cu-Zn utilizing superoxide dismutase (SOD; Fig. 4c). Different SOD proteins in diatoms employ different metal cofactors including iron[95], and the overrepresentation of Cu-Zn SOD may signify a preference for a non-iron using SOD under high $CO_2$.

Additionally, nitrogen assimilation proteins are estimated to account for a significant portion of the cellular iron requirement in phytoplankton[96], and reducing their transcript or protein abundances has also been associated with iron stress in phytoplankton[73]. Under higher $CO_2$, pennate diatoms decreased transcript expression of ammonium transporters (*AMT*) and the ferredoxin-dependent glutamate synthase (*Fd-GOGAT*) in Experiment 1 and nitrate transporters (*NRT*) in Experiment 3 (Fig. 4c and Supplementary Fig. 25). Similarly, centric diatoms decreased transcript expression of the ferredoxin-dependent nitrite reductase (*Fd-Nir*) in Experiment 3 (Supplementary Figs. 23 and 25).

At the first time point of Experiment 3, *Ostreococcus* decreased transcript expression for the entire assimilatory nitrate reduction pathway including the plastid-localized nitrite transporter (*FNT*) and the nitrate transporter accessory protein, *NAR2* (Fig. 4c). NRT and Fd-Nir also displayed significantly lower protein abundances while NAR2 abundances were not significantly different. *Fd-GOGAT* transcript expression was also lower at the first time point ($P = 0.052$, Fig. 4c and Supplementary Fig. 25), and by the second time point, protein abundances were significantly lower (Supplementary Figs. 25 and 26). *Ostreococcus* also reduced transcript expression at 800 ppm for these nitrate assimilation genes but not to the same degree as at 1200 ppm further indicating that responses are function of $CO_2$ (Supplementary

Fig. 25). Additionally, *Ostreococcus* responded by increasing transcript expression of a SLC9-family $Na^+/H^+$ exchanger at 1200 ppm, potentially to regulate intracellular pH under higher acidity (Fig. 4c)[97].

At the second time point, *Ostreococcus* displayed further iron-related responses (Supplementary Fig. 26). Like the increased protein abundances of the Cu-Zn SOD for centric diatoms, *Ostreococcus* increased expression of the Ni-using SOD. *Ostreococcus* also had reduced protein abundances of the iron storage protein ferritin (FTN) where it may play a role in buffering intracellular iron, a function that may no longer be beneficial under Fe stress[98].

In Experiment 1, centric diatoms displayed remodeling of their carbon uptake pathway with changes in carbonic anhydrase (CA) transcript and protein abundances (Fig. 4c). Carbonic anhydrases (CA) interconvert $CO_2$ and $HCO_3^-$ as part of the $CO_2$-concentrating mechanism (CCM) in diatoms to overcome relatively low concentrations of ambient $CO_2$[99]. Diatoms possess multiple CAs belonging to evolutionarily distinct families with diverse localizations including extracellular CAs at the cell surface, although there is not consistent conservation between family and location[99].

With increased $CO_2$, transcript expression of *α-CA* and *β-CA* in centric diatoms at Experiment 1 significantly increased while expression of *δ-CA* and *ι-CA* significantly decreased (Fig. 4c and Supplementary Fig. 23). These genes were not found to be differentially abundant on the protein level. Conversely, CAs in pennate diatoms at Experiment 1 were not significantly different on the transcript level, but δ-CA proteins significantly decreased with increased $CO_2$ aligning with the lower expression observed in centric diatoms.

In centric diatoms, *α-CA*, *β-CA*, and *γ-CA* expression was dominated by *Chaetoceros* whereas other centric diatom genera were expressing *δ-CA* and *ι-CA* indicating that these responses may be more genus-specific (Fig. 4c and Supplementary Fig. 23). As zinc is a common cofactor in carbonic anhydrases[100] and ZIP-family transporters may also transport zinc[79], increased expression of *ZIP1*, which was also largely from *Chaetoceros*, may serve to support increased α-CA and β-CA abundances under higher $CO_2$. α-CA and β-CA are believed to be localized to different compartments in the chloroplast in model diatoms[99], and expression of *α-CA* is thought to be constitutive[101]. γ-CAs and δ-CAs have shown a wide range of localizations[99,101] and ι-CA has been localized to the chloroplast in *T. pseudonona*[102]. Both *δ-CA* and *ι-CA* have been shown to be downregulated under high $CO_2$ as observed here, although for *δ-CA*, this pattern is not entirely consistent[101–104].

SLC4-family bicarbonate transporters are also part of the CCM[105]. Despite change in CA expression, SLC4 transporter transcripts were not differentially expressed in any experiment for centric or pennate diatoms (Supplementary Fig. 27). In the protein data, it was only detected at Experiment 1 in centric diatoms after 72 h where it was also not differentially abundant. *SLC4* expression has previously been shown to decrease with higher $CO_2$ in the model diatom *T. pseudonana*[103]; however, no difference in bicarbonate uptake or certain *SLC4* transcripts within a more extensive pH range (7.5–8.2) was observed in the model diatom *P. tricornutum* suggesting a lack of response over the pH ranges used here[105].

## Discussion

Ocean acidification is hypothesized to alter iron bioavailability[17,18], which is particularly important as large areas of the ocean including some upwelling regions experience frequent iron limitation[12–14]. Here we examined upwelling-associated phytoplankton communities' responses while they were simultaneously exposed to iron stress and acidification over short time periods. The iron status and responses of these communities were assessed with a combination of experimental, biogeochemical, physiological, and 'omics approaches. Although large differences were not observed in the bulk community, molecular-level responses with greater taxonomic resolution of dominant groups were

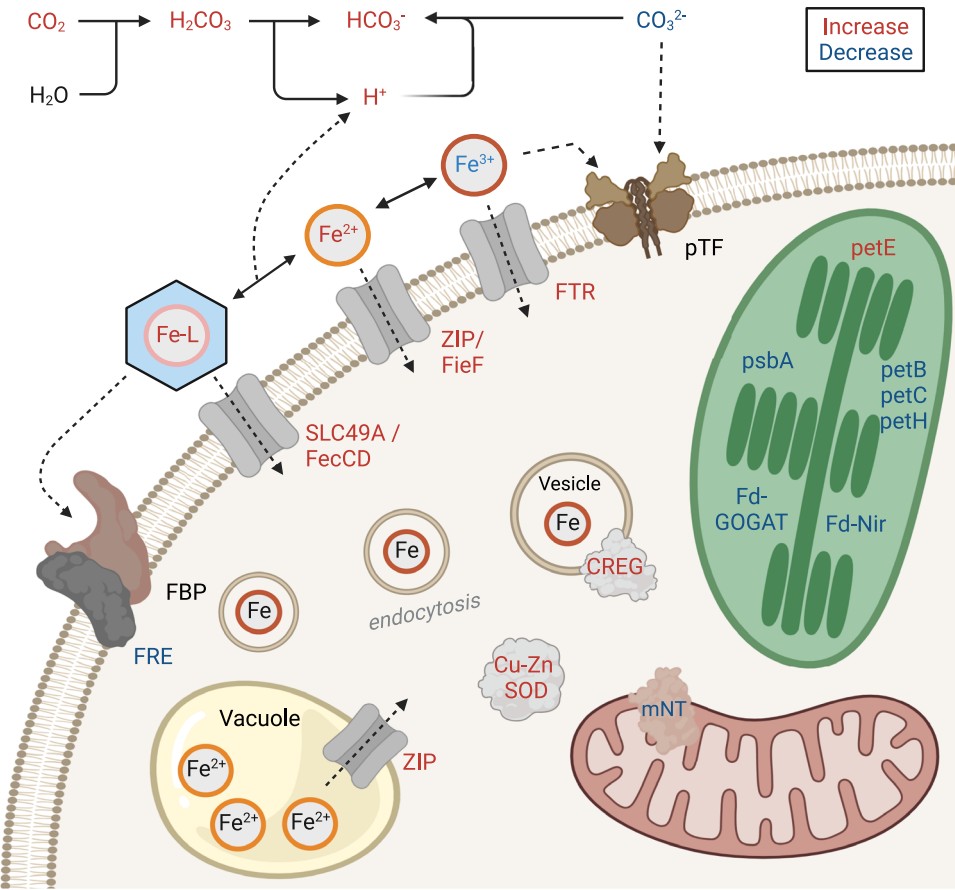

**Fig. 5 | Potential diatom-related cellular changes from increased acidification driving changes in iron chemistry and/or bioavailablity.** Molecules or proteins are colored red or blue for potential increases or decreases respectively. Black denotes no change. This figure was created with BioRender.com.

indicative of acidification-driven alterations to iron bioavailability and greater iron stress under higher $CO_2$. Within centric and pennate diatoms, these responses were variable, but included changes to myriad of genes related to iron uptake and cellular iron demand (Fig. 5). Several of the iron-related genes described here are relatively uncharacterized in diatoms and of diverse evolutionary origins highlighting diatoms' unique adaptations to grow under low iron availability.

In Experiment 1, both centric and pennate diatoms employed iron uptake pathways for siderophore-bound iron or ferrous iron that are more bioavailable under acidic conditions (Fig. 4c, 5)[17,18]. Here, a preference for siderophore-bound iron may not have been reflected in the uptake rates of Fe-DFB (Fig. 2d) as Fe-DFB uptake has been characterized with a different suite of genes than the ones observed to be upregulated here (FecCD and SLC49A). Genes responsible for Fe-DFB uptake, *FRE* and *FBP1*, were also not differentially expressed except for decreased FRE protein abundances in centric diatoms in Experiment 1, although the ferric reductases expressed here may represent separate homologs that does not serve the same functional role (Fig. 4c and Supplementary Figs. 21 and 22)[89]. Furthermore, uptake of Fe-DFB may be less affected by acidification compared to other siderophore complexes[17,18], and organically-complexed iron in seawater is generally more bioavailable[106]. In Experiment 3, the response was instead transcriptional and proteomic increases of iron permeases (FTR) in centric diatoms signifying an increase in free inorganic iron transporters as its availability is reduced (Fig. 4c). This increase in transporters is further supported by higher maximal uptake rates of inorganic iron that are also a further sign of increased iron stress (Fig. 2d).

The difference in iron transporter expression between the two experiments may have occurred for several reasons. First, the iron

status of the communities was distinctly different; the community at Experiment 1 was not iron-limited whereas the community at Experiment 3 was, although in Experiment 1, the $NO_3$:dFe ratio suggested the potential for iron limitation and high expression of ISIP genes indicates some degree of iron stress (Fig. 4b). Meanwhile, Experiment 1 also had the highest initial dFe concentrations (Fig. 1b). Collectively, this may have allowed the community at Experiment 1 to possess relatively high intracellular and/or stored iron to rely on while subjected to added iron stress from acidification. The observed increased expression of ferrous iron transporters that are potentially intracellular may support this increased reliance on intracellular recycling or release of stored iron to sustain growth over these relatively short timescales[79,81]. Second, the iron pools between the experiments were likely distinct although they are largely uncharacterized here. Besides higher dissolved iron concentrations, labile particulate iron concentrations were relatively high at Experiment 1, and dissolution of particulate iron may have supported growth (Fig. 1c and Supplementary Fig. 3). Third, the communities were distinct not only in terms of composition but also functional diversity (Fig. 3c). Only 12.2% and 21.5% of orthologous gene clusters in centric and pennate diatoms respectively were found to be shared between the two experiments (Supplementary Fig. 13). Certainly, there is high functional diversity within diatoms, and their responses may not be equivalent.

In both Experiments 1 and 3 however, diatoms also appeared to reduce their cellular iron requirements (Figs. 4c and 5, Supplementary Figs. 23 and 25). Decreased transcript or protein abundances of several iron-requiring proteins related to photosynthesis and nitrogen assimilation as well as a mitochondrial protein, mitoNEET, were observed. In Experiment 3, centric diatoms increased transcript abundances of the iron-free protein plastocyanin and increased

protein abundances of the Cu-Zn SOD which may substitute for the iron-using SOD.

Collectively, these changes in transcript and protein abundances signify an increase in iron stress resulting from acidification. The use of alternative iron uptake pathways or iron-sparing measures may have temporarily satisfied high cellular iron demand over the relatively short periods examined here, but longer-term exposure to high $CO_2$ may have resulted in a more iron-stressed community. For the community at Experiment 1, siderophore-bound iron or ferrous iron uptake pathways are likely less efficient than those for inorganic ferric iron uptake, and a reliance on iron storage is not sustainable over longer time scales. For the community at Experiment 3, the increase of iron permeases for ferric iron uptake confers increased energetic costs, and reductions in iron-requiring photosynthetic and nitrogen assimilation proteins may result in less efficiency in these critical pathways for growth. However, future iron supplies will likely be altered, and the net result remains unknown[5,107]; increased iron inputs could offset reductions in bioavailability and prevent increased iron stress.

Remodeling of carbon uptake pathways was also observed in centric diatoms in Experiment 1 (Fig. 4c and Supplementary Fig. 23). Phytoplankton employ a carbon concentrating mechanism (CCM) that includes carbonic anhydrases (CA) to overcome relatively low concentrations of ambient $CO_2$[99]. Extracellular CAs (eCA) can alter the carbonate chemistry in the region immediately around the cell[108] with activities that span orders of magnitude and scale with cell size[109]. This function may potentially overcome impacts from acidification such as maintaining elevated $\left[CO_3^{2-}\right]$ for iron uptake via phytotransferrin. In Experiment 1, certain centric diatoms decreased expression of CAs associated with low $CO_2$ as $CO_2$ increased which potentially translates into energetic savings[10]; however, *Chaetoceros* instead appeared to increase expression of α-CA and β-CA which would have the opposite effect (Fig. 4c). Potential energetic savings or costs may also be offset by other observed changes such as the relative increases in ribosomal proteins in pennate diatoms at Experiment 1 (Supplementary Fig. 16), other unknown proteins, or are negligible relative to the cell's energy demand contributing to the lack of large changes observed here.

The absence of large observable effects in Experiments 2 and 4 may be a result of reduced acidification exposure relative to community growth rates compared to Experiments 1 and 3 where multiple cell divisions may have occurred (Supplementary Fig. 8). Corresponding to this reduced exposure relative to growth rate, there were comparatively fewer transcriptional changes in Experiments 2 and 4 (Fig. 4a). Alternatively, there may have been an increase or no net change in bioavailable iron during those experiments. Under acidic conditions, the dissociation of iron from certain organic ligands is less favorable; however, this is not universal and some ligands are unaffected[17]. Under more acidic conditions, some models indicate that Fe(III) may instead be more soluble leading to increases in bioavailable dissolved iron[22,23]. Indeed, the iron pool and dynamics in the natural environment are vastly more complex than those in laboratory studies with both a single organism and iron source where reductions in iron bioavailability have been shown[19].

Within all four experiments however, short-term exposure to pCO$_2$ levels surpassing global surface seawater means predicted by the end of the century and most natural surface water variability in the CCS (Supplementary Tables 1–3) produced few observable differences in chlorophyll *a* production and nutrient uptake including that of both inorganic and organically-complexed iron (Fig. 2). Furthermore, increased $CO_2$ did not alter the carbon-to-nitrogen ratios of the community suggesting a lack of $CO_2$ fertilization with no increase in carbon content from high $CO_2$ as previously proposed (Fig. 2c)[110]. Increased acidification did not create shifts in the prokaryotic or eukaryotic microbial communities over the short duration of the experiments as there were almost no significant differences in alpha diversity or community among treatments (Fig. 3b, c). Few ASVs (<1.5%) were

estimated to be differentially abundant (Fig. 3d), and of those that were, some may be calcifying phytoplankton that were impacted as calcification became less energetically favorable[10].

These results may suggest that the natural phytoplankton communities observed here possess short-term resistance to acidification at pH values as low as 7.6 despite iron stress. Certainly, many phytoplankton taxa, especially diatoms, regularly bloom within the naturally acidic conditions of initially upwelled waters, and this resistance may have been selected for. Our experiment further offshore, Experiment 4, indicates that this lack of impact may extend farther than the nearshore region; however, these responses may still differ from regions where both pH variability and iron supply are even lower, e.g., high-nitrate low-chlorophyll regions. Again however, these results may have occurred from the relatively short duration of the experiments, and longer exposure may have showed either impacts from added iron stress or continued resistance.

Ocean acidification is in itself a multiple stressor as several carbon and iron parameters in seawater are changing simultaneously[111]. As a disturbance, it may be representative of a nonlinear transition where the tipping points are not reached until extreme conditions are encountered[112]. In the case of phytoplankton in upwelling regions, the tipping point may exist at atmospheric $CO_2$ levels beyond those estimated by the end of the century under high emissions scenarios or more prolonged exposure compared to the short time-scales used here (Supplementary Table 2). These extremes and interactive effects with additional stressors such as temperature for upwelling-associated phytoplankton communities remain to be uncovered.

## Methods

### Experimental design

Trace metal clean incubation experiments were conducted at four sites (Table 1) in August 2019 onboard the *R/V Atlantis* as part of the California Current Ecosystem Long-Term Ecological Research (CCE-LTER) Process Cruise. At each site at approximately 06:00 PST, seawater was collected from near the depth of maximum fluorescence (Supplementary Fig. 1) with a trace metal CTD-rosette fitted with Teflon coated 5 L Niskin-X bottles (Ocean Test Equipment) on a non-metallic hydroline. Niskin bottles were then brought into a positive pressure trace metal clean van where seawater was dispensed into 4 L polycarbonate bottles fitted with perfluoroalkoxy alkane (PFA) tubing and Kynar® PVDF barb fittings for sampling and bubbling. At the end of one inlet line, the bottles also had a plastic aerator (Lee's Discard-A-Stone) to maximize gas transfer.

Prior to the incubations, bottles were cleaned using trace metal techniques in a trace metal clean area under positive pressure. First, bottles were treated with acidic detergent (CitraNOX®, Alconox Inc.) for one day then rinsed five times with Milli-Q. Bottles were then filled with 1.2 mol L$^{-1}$ trace metal grade HCl for two weeks then rinsed again three times with Milli-Q. During each step, detergent, Milli-Q, or HCl was syringed through the aerator, barbs, and tubing with all plastic syringes (NORM-JECT®).

Bottles were rinsed with seawater three times, then 4 L of seawater was dispensed into each bottle. To attempt to avoid macronutrient depletion, macronutrients (nitrate, phosphate, and silicic acid) that had been prepared with trace metal grade salts and passed over Chelex 100 resin to remove contaminating metals as described by Sunda, et al.[113]. Macronutrients were added at varying concentrations based on estimated initial concentrations (Table 1). Bottles were then transferred to the laboratory onboard where they were placed in an incubator (Percival Scientific) at 12.5 °C with 115 μmol photons m$^{-2}$ s$^{-1}$ photosynthetically active radiation (PAR). The incubator was programmed on a light:dark cycle to closely match the timing of sunset and sunrise at the time and location of the incubations (lights on from 06:30 to 19:45). Seawater was then continuously bubbled with commercially prepared gas mixtures of air balanced with 400, 800, or

1200 ppm $CO_2$ (Praxair, Inc.) at a total rate of 0.3 LPM for each treatment. Gas was delivered to each bottle through a 4-port cross connector and fine-tuned with Keck™ tubing clamps to achieve evenly distributed flow of approximately 0.1 LPM in each bottle. Prior to the cross connector, gas was filtered through a 0.2 μm Millex-FG PTFE filter (Millipore), passed through tubing that also cleaned with 1.2 mol $L^{-1}$ trace metal clean HCl and rinsed three times with Milli-Q, then delivered to the inlet tubing at the bottle through another Whatman™ HEPA-Vent filter. A HEPA-Vent filter was also was placed on the outlet tubing to prevent contaminating particles from entering the bottles. Triplicate bottles were used for all experimental conditions; one replicate from the 800 ppm treatment for experiment 4 (800B) was not included in final analyses due to potential iron contamination.

Seawater was collected directly from the Niskin bottles and immediately processed for the initial time point. For the later time points, sampling occurred at one-day intervals that varied among experiments: 48 and 72 h for Experiment 1, 24 and 48 h for Experiment 2, and 48 and 96 h for Experiments 3 and 4 (Table 1). These intervals were chosen to attempt to avoid macronutrient depletion with maximum experimental duration of 96 h to also avoid increased bottle effects. To sample these later timepoints or chlorophyll *a* at 24-hour intervals, bottles were brought into a HEPA-filtered positive pressure trace metal clean area. Samples were then immediately taken for dissolved inorganic carbon (DIC) followed by subsamples for chlorophyll *a*, particulate organic carbon and nitrogen, RNA, proteins, Fe uptake, macronutrient concentrations, and total alkalinity (TA) that were dispensed into additional bottles for sampling as described below.

### Satellite data and maps
Satellite-derived data were obtained from the NOAA CoastWatch Browser, and maps were created with matplotlib v2.2.2 for Python v2.7[114]. Chlorophyll *a* concentrations on a 0.0125° grid were obtained from MODIS on board the Aqua (EOS PM).

### Carbonate chemistry
Initial (T0) seawater samples for dissolved inorganic carbon (DIC) and total alkalinity (TA) analysis were collected from the Niskin bottles. Tubing was attached to the Niskin bottle and placed at the bottom of a 250 mL borosilicate glass bottle. The bottle was then inverted to thoroughly rinse the inside of the bottle and then filled from the bottom and allowed to overflow at least one full volume. The tubing was then pinched and withdrawn leaving only a small headspace. 100 μL of a saturated mercuric chloride ($HgCl_2$) solution was added to the sample to prevent biological activity[114]. Bottles were then sealed with high vacuum grease (Apiezon) on the stopper and fastened using a rubber band and bottle clip.

For the later timepoints, DIC and TA samples were collected separately. DIC was collected immediately by slowly syringing without introducing bubbles from the incubation bottles into 60- or 120-mL borosilicate glass bottles. Each bottle was gently rinsed with seawater then filled from the bottom using tubing connected to the syringe. Seawater was immediately poisoned with either 25 μL or 50 μL of saturated $HgCl_2$ solution for the 60- or 120-mL bottles respectively. Bottles were then sealed with grease and clipped as done for the T0 samples. DIC was measured with an Automated Infra-Red Inorganic Carbon Analyzer (AIRICA, Marianda) equipped with a LI-COR 7000 $CO_2/H_2O$ gas analyser as the detector.

TA was collected from filtrate from the sterivex used for RNA or proteins into 250 mL borosilicate glass bottles. Each bottle was rinsed thoroughly three times before filling. Seawater was poisoned with 100 μL saturated $HgCl_2$ solution and the bottles were sealed as described for the T0 samples. TA was then measured with an open-cell potentiometric acid titration (0.1 N HCl) system developed by the Dickson Lab at Scripps Institution of Oceanography[115].

Both seawater DIC and TA measurements were made relative to certified reference materials (CRMs) provided by the laboratory of A. Dickson at Scripps Institution of Oceanography. The accuracy and precision of the measurements from the CRMs were -0.21 ± 1.26 μmol $kg^{-1}$ ($n = 24$) for DIC and -0.42 ± 1.85 μmol $kg^{-1}$ ($n = 17$) for TA. The complete seawater carbonic-acid system (i.e., pCO2, pH, $CO_3^{2-}$, $HCO_3^-$ etc) was calculated based on DIC, TA, temperature, salinity, phosphate, and silicate using CO2SYS for MATLAB (v 2.1)[116]. Calculations used the first and second dissociation constants of carbonic acid ($K_1$ and $K_2$) from Mehrbach, et al.[117] refit by Dickson and Millero[118], the dissociation constants of bisulfate ($K_{HSO4}$) from Dickson[119], and total boron from Uppstrom[120]. pH values are reported on the total $H^+$ scale.

### Dissolved iron concentrations
Dissolved iron (dFe) was measured via chemiluminescence flow-injection analysis with hydrogen peroxide oxidation (FeLume, Waterville Analytical) as described by Lohan, et al.[121]. Briefly, seawater was filtered using acid-cleaned 0.2 μm Acropak 200 capsule filters (VWR International) into clean and rinsed low-density polyethylene (Nalgene) bottles in a positive pressure clean van then was acidified to pH ~ 1.8 with HCl (Optima grade, Fisher Scientific). Dissolved iron was oxidized to Fe(III) with hydrogen peroxide (1% v/v optima grade), buffered in-line with an ammonium-acetate mixture, and selectively preconcentrated on a column packed with resin (Toyopearl 650 M chelating resin) at pH ~3.5. After the sample was eluted with 0.23 M HCl (Optima grade), the production of radicals catalyses a three-step oxidation reaction with hydrogen peroxide (0.23 M optima grade) and luminol (0.25 mM). Iron was then eluted and oxidized with pH > 9 luminol-ammonia buffer, and the chemiluminescence (425 nm) was measured with a photomultiplier tube. Final dissolved iron concentrations were then quantified via external standard curve. In-house and GEOTRACES consensus reference standards (geotraces.org) were measured in each analytical run to ensure accuracy and precision.

### Labile particulate iron concentrations
Labile particulate iron was calculated as total dissolvable Fe (TDFe) minus dissolved iron (dFe). Unfiltered samples were acidified to pH ~ 1.8 with HCl (Optima grade, Fisher Scientific) and stored for 22–24 months. Immediately prior to analysis, the samples were syringe-filtered using acid-cleaned syringes and 0.2 μm Supor® filters, and then analyzed using the same analytical methods as for dFe samples. The blank from the syringe-filtration was averaged 0.058 nmol $L^{-1}$, which is at most a 6% correction to the overall measurement of TDFe. Due to the calibration range of the standards, it was necessary to dilute many of the samples, especially nearshore; therefore, offshore filtered Pacific surface seawater (0.22 nmol $L^{-1}$) was used to dilute samples by a factor of 10 prior to analysis.

### Dissolved macronutrient concentrations
Approximately 30-35 mL of filtrate from the Sterivex filter unit (0.2 μm) was dispensed into 50 mL polypropylene centrifuge tubes and frozen at -20 °C until analysis. Each tube was rinsed three times before filling. Nutrients were analysed on a QuAATro continuous segmented flow autoanalyzer (SEAL Analytical) as described by the CalCOFI methods manual. Total oxidized nitrogen (nitrate reduced to nitrite), solely nitrite, and the efficiency of reduction of nitrate to nitrite were assessed to calculate solely the nitrate concentration. Reference materials for nutrients in seawater (KANSO Co., LTD.) were included in the run for quality control.

### Si$_{ex}$ proxy for diatom iron limitation
Si$_{ex}$ is a biogeochemical proxy that suggests Fe limitation in diatoms due to preferential utilization of silicic acid[53,55,122], and was calculated as described by Hogle, et al.[53]. Briefly, $Si_{ex} = \left[ \mu mol\, H_4 SiO_4\, L^{-1} \right] - (\left[ \mu mol\, NO_3\, L^{-1} \right] * R_{Si:NO3})$ where $R_{Si:NO3}$ is the preformed ratio of silicic

acid and nitrate from upwelled source waters. Based on averaged values from the upwelling source water densities in the region ($\sigma_\theta = 25.8\,kg\,m^{-3}$), $R_{Si:NO3} = 0.99$[53].

## Parallel incubations to assess iron status

Parallel incubations were conducted to evaluate the presence of iron limitations. First, approximately four hours prior to the start of each of the $pCO_2$ experiments, seawater was collected from near the same depth as the $pCO_2$ experiments with the TMC rosette (Supplementary Fig. 1). One litre of seawater was dispensed into trace metal clean polycarbonate bottles. Trace metal clean techniques were used as described above. Triplicate bottles at each site were either unamended (control), supplemented with 5 nmol L$^{-1}$ iron (+Fe), or supplemented with 100 nmol L$^{-1}$ of the iron chelator desferrioxamine B (+DFB).

Bottles were incubated in situ for 24 h with a quasi-Lagrangian drifter as described in Stukel, et al.[123]. The drifter was equipped with a surface float and holey-sock drogue centered at 15 m. Bottles were attached to the drifter near the depth of collection in mesh bags. Either immediately (T0) or after 24 h of incubation, samples were collected for chlorophyll $a$ and macronutrients. The drifter was followed between the start of these incubations and the acidification incubations to sample from approximately the same water mass.

One day following the initiation of the fourth $pCO_2$ experiment (E4) and while still following the Lagrangian drifter, an additional trace metal clean incubation experiment was performed using a community collected and incubated on the drifter at 40 m depth. Triplicate 1 L polycarbonate bottles at each site were either unamended (control), supplemented with 5 nmol L$^{-1}$ (+Fe), supplemented with 10 μmol L$^{-1}$ (+N), or both iron and nitrate (+FeN). After 48 h, samples were collected for chlorophyll $a$.

In addition to the incubations at E3 on the drifter, deckboard incubations in triplicate 2.7 L polycarbonate bottles were performed with neutral density screening for light and flow-through seawater for temperature. These bottles were either unamended (control) or supplemented with 5 nmol L$^{-1}$ (+Fe). Samples for chlorophyll $a$ were collected immediately (T0) and following 24 and 48 h of incubation.

## Chlorophyll $a$

Fifty mL of seawater was filtered through a GF/B filter (1.0 μm, 25 mm) for the acidification experiments or a GF/F filter for the parallel incubations under gentle vacuum pressure (<100 mm Hg) and frozen at −20 °C until analysis. Chlorophyll $a$ extraction was performed using 90% acetone at −20 °C for 24 h and measured via in vitro fluorometry on a Turner Designs 10-AU fluorometer using the acidification method. Community growth rates (μ) were derived from the slope of linear regressions with the natural log of chlorophyll $a$ concentrations over time. The last time point from Experiment 1 was excluded from the growth rate calculations as macronutrients were depleted.

## Particulate organic carbon and nitrogen

Four hundred mL of seawater was filtered through a pre-combusted Whatman GF/B filter (1.0 μm, 25 mm, 450 °C for 6 h) under gentle vacuum pressure (<100 mm Hg) and frozen in plastic petri dishes at −20. Filters were dried at 62 °C then inorganic carbon was removed by acidifying the filters with HCl vapor followed by drying for another hour at 62 °C. Filters were then encapsulated in tin and analysed on a Costech ECS 4010 CHNSO Analyzer following the manufacturer's instructions. Blank filters were run alongside the samples and subtracted from the sample quantities.

## Short-term $^{59}$Fe uptake assays

Iron uptake rates were characterized in subsamples collected from the incubation bottles or directly from the Niskin-X bottles (T0 samples) in acid-cleaned polycarbonate bottles. Additions of either 1 nmol L$^{-1}$ $^{59}$FeCl$_3$ or 1 nmol L$^{-1}$ $^{59}$Fe-DFB (ferrioxamine B) were made to uptake bottles which were then incubated without light. $^{59}$Fe-DFB stocks were equilibrated overnight with a 4:5 ratio of iron to DFB. Four aliquots of each uptake bottle were filtered onto 1 μm polycarbonate filters at one-hour intervals. Each filter was rinsed with an oxalate-EDTA wash solution for 15 min before rinsing with filtered seawater[124]. Filters were then preserved in Ecolite™ liquid scintillation cocktail (MP Biomedicals), and $^{59}$Fe was quantified by liquid scintillation counting. Uptake rates were calculated by the linear regression of particulate $^{59}$Fe per litre of seawater against time. Rates were normalized to both chlorophyll $a$ and particulate organic carbon concentrations.

## RNA collection and extraction

Between 525 and 1100 mL of seawater from each bottle or the initial seawater was filtered through a 0.22 μm Sterivex-GP filter unit (Cat. no. SVGP01050, Millipore) which was immediately sealed with a 1.2 mol L$^{-1}$ HCl and Milli-Q rinsed luer-lock plug and Hemato-Seal™ tube sealant, wrapped in aluminum foil, and flash frozen in liquid nitrogen. RNA was extracted using the NucleoMag RNA extraction kit (Machery-Nagel) on an Eppendorf epMotion 5075t as described here: https://doi.org/10.17504/protocols.io.bd9ti96n. RNA quality and quantity were assessed with an Agilent 2200 TapeStation (RNA ScreenTape) and the Quant-iT™ Ribogreen® RNA Assay kit. For RNA at initial time point (T0) for Experiment 3, $n = 2$.

## 16S and 18S rRNA amplicon sequencing and analysis

From the aforementioned extracted RNA, cDNA was synthesized with the Invitrogen SuperScript™ III First-Strand Synthesis System according to the manufacturer's instructions. RNA-derived amplicon libraries targeting the V4-V5 hypervariable region of the 16S gene and the V9 hypervariable region of the 18S region were then generated via a one-step PCR using the cDNA with the Azura TruFi DNA Polymerase PCR kit. For 16S, the 515F-Y (GTGYCAGCMGCCGCGGTAA) and 926R (CCGYCAATTYMTTTRAGTTT) primer set was used[125]. For 18S, the 1389F (TTGTACACACCGCCC) and 1510R (CCTTCYGCAGGTTCACCTAC) primer set was used[126]. Each reaction was performed with an initial denaturing step at 95 °C for 1 min followed by 30 cycles of 95 °C for 15 s, 56 °C for 15 s, and 72 °C for 30 s. 2.5 μL of each PCR reaction was ran on a 1.8% agarose gel confirm amplification. PCR products were purified using Beckman Coulter AMPure XP following the standard PCR clean-up protocol. PCR quantification was performed using Invitrogen Quant-iT PicoGreen dsDNA Assay kit. Samples were then pooled in equal proportions (10 ng DNA per sample) followed by another AMPure XP PCR purification. The final libraries were evaluated on an Agilent 2200 TapeStation (D1000 ScreenTape), quantified with Qubit HS dsDNA kit, and each sequenced at the University of California, Davis Sequencing Core on a single Illumina MiSeq lane with a 15% PhiX spike-in.

Amplicons were analysed with QIIME2 v2019.10[127]. Briefly, demultiplexed paired-end reads were trimmed to remove adapter and primer sequences with cutadapt[128]. Trimmed reads were then denoised with DADA2 to produce amplicon sequence variants (ASVs)[129]. Taxonomic annotation of ASVs was conducted with the q2-feature-classifier classify-sklearn naïve-bayes classifier[130,131] and the SILVA database (Release 138) for 16S[132] or the PR$^2$ database (v4.12.0) for 18S[133]. 16S ASVs with eukaryotic, plastid, and mitochondrial taxonomic assignments were removed. Diversity metrics and significance tests were calculated with the QIIME2 diversity plugin with samples rarefied to 3,756 reads for the 16S data and 82,249 reads for the 18S data (Supplementary Fig. 28)[134,135]. Differential abundances of ASVs were assessed using the final time points of each experiment with DESeq2 v1.22.2[136–138]. ASVs were considered differentially abundant by those with Benjamini and Hochberg adjusted $P$-values < 0.05[139].

## Metatranscriptome library preparation, sequencing, and analysis

A poly-A selected metatranscriptome sequencing library was constructed with the Illumina TruSeq Stranded mRNA kit according to the manufacturer's instructions and sequenced on an Illumina NovaSeq 6000 with a S4 flow cell (2 x 150 bp). Raw reads were quality trimmed with trimmomatic v0.39[140] and evaluated with FastQC v0.11.8 and MultiQC v1.9.dev0[141]. Trimmed reads were assembled into contigs utilizing subsets of samples to generate sub-assemblies with Trinity v2.8.5 with *min_contig_length* set to 150 bases[142]. These assemblies were then merged into final assemblies for each experiment with Trans-AbySS v2.0.1[143].

Proteins were predicted from the final assemblies with GenemarkS-T[144] which were deduplicated using the nucleotides of the predicted proteins and the dedupe script from BBMap v38.79. Ribosomal RNA sequences were identified with SortMeRNA v2.1 and were also removed[145]. Reads were mapped to the predicted proteins with Bowtie2 v2.3.5 in local alignment mode[146]. Only reads with a MAPQ score $\geq 8$ were retained. Taxonomic annotation of predicted proteins was performed with DIAMOND (v0.9.31) BLASTP searches against PhyloDB v1.076[147,148] with an E-value cutoff of $10^{-5}$, minimum query coverage of 60%, and alignment sores in the top 95%. To avoid misannotations from contamination in the reference database, final taxonomic assignment was based on the Lineage Probability Index from the BLASTP hits with the aforementioned cutoffs[148,149]. Orthogroups across different experiments were identified using the hierarchial phylogenetic approach implemented in OrthoFinder v2.5.4[150]. For functional annotation, DIAMOND BLASTP searches were performed against the Kyoto Encyclopedia of Genes and Genomes (KEGG; Release 94.1)[148,151]. KEGG Ortholog (KO) assignment was performed with KofamKOALA which utilizes hmmsearch against KOfam, an HMM database of KOs[152]. Within taxonomic groups, read counts were summed by KO annotation, and if KO annotation was absent, the KEGG gene annotation was used. Normalization and differential expression of transcripts were then assessed with DESeq2 v1.22.2. Significance in differentially expressed transcripts was determined with Benjamini and Hochberg adjusted *P*-values (P < 0.05)[139]. MA plots were generated with the code provided in Cohen, et al.[153].

## Protein sample collection, extraction, and preparation

Seawater (300-900 mL) was filtered through a 0.22 μm Sterivex-GP filter unit (Cat. no. SVGP01050, Millipore) which was immediately sealed with a 1.2 mol L$^{-1}$ HCl and Milli-Q rinsed luer-lock plug and Hemato-Seal™ tube sealant, wrapped in aluminum foil, and flash frozen in liquid nitrogen. Proteins were extracted with a detergent-based procedure based on Saito, et al.[154]. First, filters were removed from the Sterivex casing as described by Cruaud, et al.[155] and placed in 750 μL of extraction buffer (2% SDS, 0.1 M Tris/HCl pH 7.5, 5 mM EDTA). Samples were incubated for 10 min at 95 °C while mixed at 350 rpm. Sonication was then performed with a Diagenode Bioruptor for approximately 8 min (15 s on/off intervals) at 4 °C. Next, samples were again incubated at room temperature for 30 min while gently vortexing every 10 min. The filters were then removed from the tubes, and the remaining liquid was centrifuged at 15,000 × *g* for 30 min at room temperature. The supernatant containing the extracted protein was then removed without disturbing the pellet. Crude protein extracts were quantified with the Pierce™ colorimetric BCA protein concentration assay.

S-Trap™ mini columns (PROTIFI) were then used to remove SDS extraction buffer from the samples and to digest extracted proteins into mass-spectrometry-compatible peptides. Proteins were first reduced and alkylated using 5 mmol L$^{-1}$ dithiothreitol and 15 mmol L$^{-1}$ iodoacetamide respectively, and then denatured using 12% phosphoric acid. Samples were then diluted (1:7) with S-trap™ buffer (pH 7.1 100 mmol L$^{-1}$ tetraethylammonium bicarbonate in 90% aqueous

methanol) and loaded onto the S-trap™ columns. To remove SDS, each sample was washed 10 times with 600 μL S-trap™ buffer using gentle vacuum. Proteins were then digested on-column using trypsin (1 trypsin: 25 protein) at 37 °C for 16 h. Tryptic peptides were eluted from the S-Trap™ columns with a series of 50 mmol L$^{-1}$ ammonium bicarbonate, 0.2% aqueous formic acid, and 50% acetonitrile + 0.2% formic acid washes. Eluted samples were desalted on pre-conditioned 50 mg C18 columns (HyperSep), dried using a Vacufuge Plus (Eppendorf) and then resuspended in a 1% formic acid, 3% acetonitrile solution prior to liquid chromatography – tandem mass spectrometry analysis.

## Metaproteomic processing and analysis

Samples were analyzed on a Dionex UltiMate™ 3000 RSLC-nano LC system (Thermo Scientific) coupled to a Q-Exactive hybrid quadrupole-Orbitrap mass spectrometer (Thermo Scientific). Each sample was analyzed in duplicate, and a blank was run between each set of biological replicates. The peptide matrix was separated using one-dimensional liquid chromatography for 125 min under a nonlinear gradient (Supplementary Table 6). The mass spectrometer was operated under positive polarity, data-dependent acquisition mode. Detailed mass spectrometer settings are shown in (Supplementary Table 7).

Raw mass spectrometry files were converted to mzML format using ThermoRawFileParser[156]. Two annotated databases were compiled using experiment-specific metatranscriptome sequences (see *Metatranscriptome library preparation, sequencing, and analysis*) and appended with proteins from the common Repository of Adventitious Proteins to detect contaminants. Databases were then filtered to remove redundant sequences, and database searching was conducted using OpenMS MSGF+ with a 1% false discovery rate[157,158]. Post translational modification parameters included cysteine carbamidomethylation as a fixed modification, and methionine oxidation and N-terminal glutamate to pyroglutamate conversion as variable modifications.

Peptides were quantified using MS1 ion intensities, and Feature-FinderIdentification was used to identify unknown features based on identified MS2 spectra in replicate injections[159]. Individual peptide abundances in each injection were normalized to the sum of all peptide intensities in that injection (total sum scaling), excluding peptides matching to the common contaminants database. Unique peptides were assigned to a taxonomic and functional annotation based on the metatranscriptome assembly and annotation. Non-unique peptides, i.e., peptides matching to more than one taxonomic group or functional annotation, were considered to be ambiguous. As with the transcripts, peptides were grouped based on KEGG Ortholog (KO) and gene annotations within each taxonomic group. Significant differences in protein group abundance for each taxonomic group were assessed within experiments using ion intensities with DESeq2 and Benjamini and Hochberg adjusted *P*-values (P < 0.05)[136,139]. Although DESeq2 is designed for RNA-seq data, here it was used for consistency between the transcript and protein analyses, and it has been found to be effective at determining significance in differentially expressed proteins[160].

Ribosomal and photosynthetic protein mass fractions were calculated as described by McCain, et al.[75]. Specifically, KO gene definitions were searched against character strings for each function. The photosynthesis group included all proteins with "photosyn*", "light-harvesting complex", "flavodoxin", or "plastocyanin" in their annotation where "*" represents a wildcard character. The ribosome group included all proteins with "ribosom*" and excluded proteins responsible for ribosomal synthesis. To be included, each taxonomic group of interest was required to have more than 50 unique peptides for each protein group to avoid known biases[75]. Mass fractions were then calculated by summing the peptide abundances for each function within each taxonomic group and then dividing by the total peptide

abundancefor the taxonomic group. Differences in proteomic mass fractions were evaluated via ANOVAs followed by Tukey's honest significant difference test in R.

## Phylogentic analyses

Taxonomically representative sets of homologs of diatom FTR, CDF-family, FecCD-domain, SLC49-family, and mitoNEET genes with an emphasis on unicellular eukaryotic algae were identified from the NCBI nr and MMETSP databases via BLASTP searches (E-value $\leq 10^{-3}$)[161,162]. To reduce the complexity stemming from the transcriptomes in MMETSP, sequences were clustered at the 95% similarity level with USEARCH[163]. Seed sequences were then aligned using MAFFT v7[164]. Poorly aligned regions and regions rich in gaps were identified and removed using trimAI in *gappyout* mode[165]. Alignments were then visualized and inspected in SeaView 4[166]. Maximum likelihood phylogeny was computed in IQTree 2 under the model that produced the best fit using ModelFinder[167,168]. Ultra-fast bootstrap approximation was estimated from 10,000 replicates.

## Statistical analyses

Two-way ANOVAs followed by Tukey's honest significant difference test were performed on the biological and chemical properties of the seawater using Graphpad PRISM v9.0.0.

## Reporting summary

Further information on research design is available in the Nature Portfolio Reporting Summary linked to this article.

## Data availability

The sequence data for the metatranscriptome and rRNA amplicon libraries reported in this study have been deposited in the National Center for Biotechnology (NCBI) sequence read archive under the BioProject accession no. PRJNA787648. Metatranscriptome assemblies, read counts, and annotations are available at Zenodo [https://doi.org/10.5281/zenodo.5758778]. The ASV tables with abundances and taxonomic annotations are available as Supplementary Datasets S1 and S2 for 18S and 16S rRNA respectively. The mass spectrometry proteomics data have been deposited to the ProteomeXchange Consortium via the PRIDE partner repository with the dataset identifier PXD038549[169]. Phylogenetic trees are available via Figshare [https://doi.org/10.6084/m9.figshare.22589218]. The SILVA database was obtained from the SILVA website [https://www.arb-silva.de/], and the PR2 database was obtained from GitHub [https://github.com/pr2database/pr2database/]. The PhyloDB database is available on the A.E. Allen Lab website [https://allenlab.ucsd.edu/data/].

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

## Acknowledgements

We thank the captain and crew of the *R/V Atlantis* as well as the participants of the California Current Ecosystem Long-Term Ecological Research (CCE LTER) P1908 Cruise (AT42-15). In particular, we thank Mark Ohman, Mike Stukel, and Ralf Goericke for their contributions to the overall cruise operations and Max Fenton who assisted with sample collection. We also thank Jeff McQuaid for advice on the experimental design. This work was supported by the National Science Foundation Grants OCE-1756884 (to A.E.A.), OCE-1637632 (to K.A.B.), OCE-1756860 (to K.A.B. and A.J.A.), OCE-2224726 (to K.A.B. and A.E.A.), and DGE-1650112 (Graduate Research Fellowship to R.H.L), the National Ocean and Atmospheric Administration Grant NA19NOS4780181 (to A.E.A.), the Simons Foundation Collaboration on Principles of Microbial Ecosystems (PriME) Grant 970820 (to A.E.A.), the Simons Foundation Grant 504183 (to E.M.B.), the National Sciences and Engineering Research Council of Canada Grant RGPIN-2015-05009 (to E.M.B.), the Czech Science Foundation Grant 21-03224S (to M.O.), and the European Regional Development Fund ERDF/ESF Grant CZ.02.1.01/0.0/0.0/16_019/0000759. R.H.L. was also partially supported by a Chateaubriand Fellowship from the Office of Science and Technology of the Embassy of France in the United States and Inserm. This work also used the Extreme Science and Engineering Discovery Environment (XSEDE), which is supported by National Science Foundation grant ACI-1548562. Specifically, the Comet cluster at the San Diego Supercomputer Center was used through allocation TG-OCE200007 (to R.H.L.).

## Author contributions

R.H.L, T.H.C., K.O.F., S.K., E.M.B., A.J.A., K.A.B., A.E.A. designed the research; R.H.L., T.H.C., K.O.F., L.J.J, S.K., A.H., M.O., A.J.R., E.R., H.Z., K.A.B. performed the research; R.H.L., T.H.C., K.O.F., L.J.J, S.K., E.M.B., A.H., M.O., K.A.B., A.E.A. analysed the data; R.H.L. drafted the manuscript; all authors contributed to the discussion of results and preparation of the manuscript.

## Competing interests

The authors declare no competing interests.
