## [Peer Review File · Nature Communications]

Short-term acidification promotes diverse iron acquisition and conservation mechanisms in upwelling-associated phytoplanktonREVIEWER COMMENTS

Reviewer #1 (Remarks to the Author):

Review to 'Diverse iron acquisition and conservation mechanisms support resistance to acidification in upwelling-associated phytoplankton' by Lampe et al.

General comments

This study assessed the effect of ocean acidification (OA; 400 vs 800 and 1200 μatm pCO_2) via the performance of 4 iron-addition phytoplankton bottle incubation experiments along the Californian upwelling system. The latter enabled to assess OA effects under different degrees of iron limitation and with differently aged upwelled waters. Newly upwelled waters were investigated via experiment 1, experiments 2 and 3 tested progressively aged waters while experiment 4 waters were sampled offsite. In comparison, initial communities of experiment 1 and 2 were iron-limited while the other start communities were not. The duration of the experiments varied, spanning between 2 (experiment 2), 3 (experiment 1) up to 4 days (experiment 3,4). In my opinion, this study suffers from insufficient incubation time for exp. 1 and 2, which caused no apparent physiological, transcriptional, proteomic and community differences. This is in particular a pity, as in particular experiments 1 and 2 were the ones that only allowed to assess the effect of iron limitation next to OA, but they unfortunately also had the shortest incubation time. I agree with the authors that the assessment of transcriptomic responses allows already to assess subtle changes within short times (2 days), but then still the question remains whether these responses will be also reflected in phytoplankton growth and biomass build up (chlorophyll, POC). To see these responses at least a duration of 4 days is needed. Indeed, the authors point out "there were no observable differences in chlorophyll a production and nutrient uptake" (L371) as well as "increased CO_2 did not alter the carbon-to-nitrogen ratios of the community suggesting a lack of CO_2 fertilization with no increase in carbon content from high CO_2 " (L373-374). Unfortunately, I have serious doubts that the overall conclusion drawn from experiments 1 and 2 for upwelled and iron-limited waters is mainly an artefact of the experimental design and as such a result of the short incubation time of experiment 1 and 2 (experiment 1 = 3 days, experiment 2 = 2 days).

I agree with the authors that molecular data on iron uptake are interesting and provide interesting insights about the resistance of phytoplankton to OA over a short time scales (between 2 up to max. 4 days), but I disagree that these data will allow to confer the long-term response, as the experimental duration of only 2 days did not even allow the phytoplankton cells to adjust their physiological machinery to the applied different environmental conditions (acidification, iron). Hence, in my opinion, the presented data for experiment 1 and 2 do not even allow to draw any conclusions on the long-term response. I suggest the authors to mainly focus on the short-term responses identified at the molecular level on iron uptake and to avoid to draw any conclusion on 'chronic long-term exposure' (L52) because the authors do not even have data to make predictions on the capacity of phytoplankton on a longer term. Overall, I disagree that the manuscript currently as it is can be published in Nature Communications and suggest the authors to rewrite the manuscript by concentrating solely on the short-term responses at the molecular level on iron and carbon uptake.

Abstract:

There is large emphasis put on the fact that OA was investigated in iron-stressed phytoplankton of upwelled waters (L44-46), but as laid out by me before, iron-stressed upwelled phytoplankton was mainly found at the sampling sites of experiment 1 and 2. The latter two experiments, however, suffered from a too short incubation time (2 or 3 days), which avoids to obtain at all potential shifts in phytoplankton growth, nutrient uptake and community composition. From these experiments, however, important conclusions are drawn that “phytoplankton growth, nutrient uptake, and community compositions remained unchanged” (L47-48), which indicates their potential resistance to OA. I agree with the authors that ‘molecular-level responses suggest that resistance to acidification-driven changes in iron bioavailability is facilitated by iron uptake pathways that are less hindered by acidification and strategies that reduce cellular iron demand.’ Therefore, I strongly suggest the authors to rewrite this manuscript with a focus solely on the short-term response at the molecular level on the different iron uptake strategies. Please note that the authors even point out by themselves that for experiment 2 “insufficient incubation time may not have led to community differences” (L216-217) detected with DESeq2.

Introduction:

L93-100: I disagree with the authors that “A relatively small number of field-based studies have examined the effects of ocean acidification on iron stress in phytoplankton”. In fact, there are various studies that have performed Fe-enrichment experiments simulating OA conditions in low Fe regions such as Shi et al. 2010 Science, Feng et al. 2011 Deep Sea Res, Sugie et al. 2013 Biogeosciences, Endo et al. 2013 JEMBE, Yoshimura et al. 2013 J Oceanogr, Yoshimura et al. 2014 Deep-Sea Res, Trimborn et al. 2017 MEPS, Pausch et al. 2022 Frontiers Mar Sci. Moreover, these studies did not add any chelating agent to induce Fe stress, therefore this section needs to be rephrased.

L113-115: The authors point out that “physiological measurements may not be sensitive enough to detect more subtle change, particularly over relatively short time scales”. To me, it is not really clear why a short-term response is more of relevance than a long-term response. In fact, it is much more of interest to better understand how well microorganisms are adapted on a long-term (up to weeks) to upwelling and low Fe conditions than on a short-term (1-4 days). Moreover, there are also physiological adjustments by microorganisms that can immediately respond, as an example the photochemical quantum yield, commonly used as an important indicator of the physiological status of phytoplankton, can already change within 24h. Also, carbon uptake characteristics (CA, CO₂ and HCO₃⁻ uptake) can adjust very quickly. Hence, in my opinion the statement that molecular tools per se are better than physiological measurements is simply not appropriate.

Discussion:

L367-381: As mentioned above, the observations on growth, community composition and biomass formation in experiments 1 and 2 cannot be considered due to their too short incubation time.

L388-294: Measurements of internal and external carbonic anhydrase activities for various phytoplankton groups and species exist and it would be good to take them in the discussion into account.

L398-420: Different Fe uptake strategies were observed for the communities of experiment 1 and 3, this

is somewhat addressed, but not really in detail. This is also due to the fact that the community of experiment 1 was Fe-limited already at the start of the experiment, whereas the community of experiment 3 only became Fe-limited due to the addition of macronutrients. I think this is important to note, but this fact is completely neglected here in the discussion.

Reviewer #2 (Remarks to the Author):

Lampe et al examine the responses of diatoms to coastal upwelled waters that were experimentally acidified by increased CO₂ concentrations. The goal of the study is to see how the phytoplankton community structure and physiology responds. They hypothesize that since coastal upwelling regions experience increased pH variability in comparison to open ocean regions, the microbial communities may be adapted to handle changes in acidity and associated iron stress. The results fit with this hypothesis and there are largely no significant differences in physiological or growth conditions between the three CO₂ conditions (400, 800, 1200 ppm).

The paper is well written and clearly communicates the results. The science is sound, using well established methods in phytoplankton ecophysiology and molecular methods. The methods are adequate to allow replication and verification of the study.

I applaud this paper for being upfront about negative results. For the most part acidification leads to few or no differences in phytoplankton physiology or induce iron limitation due to acidification. This is true even though macronutrients were added to all conditions to try and push the system into iron limitation. My major concern is that it is difficult to comprehend the robustness of their metatranscriptome results across the four different experiments and conditions. The authors state up front that the metatranscriptome results were variable, and chose to present a handful of examples from select organisms within select time points from select treatments in select experiments. Figure 4 shows this best. To determine the robustness of the transcriptional trends presented in the main text, it would be helpful if the authors provided an index of what proportion of treatments, conditions, and experiments a particular gene was found to be significantly differentially transcribed.

A few detailed comments:

The proteomics data seems like a side note in the manuscript and under utilized.

Figures 1 and 2 were cutoff in the pdf.

Figure 3A and B could use some clarification. Is this just all the samples from the four time points combined into a single plot for each experiment? If so, why show the data this way which obscures the temporal trends in community composition.

Ln 196. You mention that SAR86 is the prevalent member of the gamma proteobacteria. For the alphaproteobacteria did you see SAR11 was most abundant? I ask given I was surprised to see that the gammas are so much more abundant than the alphas in fig 3 and I'm wondering if the primer sets chosen are biased against the SAR11 clade members?

Reviewer #1 (Remarks to the Author):

Review to 'Diverse iron acquisition and conservation mechanisms support resistance to acidification in upwelling-associated phytoplankton' by Lampe et al.

General comments

This study assessed the effect of ocean acidification (OA; 400 vs 800 and 1200 $\mu\text{atm pCO}_2$) via the performance of 4 iron-addition phytoplankton bottle incubation experiments along the Californian upwelling system. The latter enabled to assess OA effects under different degrees of iron limitation and with differently aged upwelled waters. Newly upwelled waters were investigated via experiment 1, experiments 2 and 3 tested progressively aged waters while experiment 4 waters were sampled offsite. In comparison, initial communities of experiment 1 and 2 were iron-limited while the other start communities were not. The duration of the experiments varied, spanning between 2 (experiment 2), 3 (experiment 1) up to 4 days (experiment 3,4). In my opinion, this study suffers from insufficient incubation time for exp. 1 and 2, which caused no apparent physiological, transcriptional, proteomic and community differences. This is in particular a pity, as in particular experiments 1 and 2 were the ones that only allowed to assess the effect of iron limitation next to OA, but they unfortunately also had the shortest incubation time. I agree with the authors that the assessment of transcriptomic responses allows already to assess subtle changes within short times (2 days), but then still the question remains whether these responses will be also reflected in phytoplankton growth and biomass build up (chlorophyll, POC). To see these responses at least a duration of 4 days is needed. Indeed, the authors point out "there were no observable differences in chlorophyll a production and nutrient uptake" (L371) as well as "increased CO₂ did not alter the carbon-to-nitrogen ratios of the community suggesting a lack of CO₂ fertilization with no increase in carbon content from high CO₂" (L373-374). Unfortunately, I have serious doubts that the overall conclusion drawn from experiments 1 and 2 for upwelled and iron-limited waters is mainly an artefact of the experimental design and as such a result of the short incubation time of experiment 1 and 2 (experiment 1 = 3 days, experiment 2 = 2 days).

I agree with the authors that molecular data on iron uptake are interesting and provide interesting insights about the resistance of phytoplankton to OA over a short time scales (between 2 up to max. 4 days), but I disagree that these data will allow to confer the long-term response, as the experimental duration of only 2 days did not even allow the phytoplankton cells to adjust their physiological machinery to the applied different environmental conditions (acidification, iron). Hence, in my opinion, the presented data for experiment 1 and 2 do not even allow to draw any conclusions on the long-term response. I suggest the authors to mainly focus on the short-term responses identified at the molecular level on iron uptake and to avoid to draw any conclusion on 'chronic long-term exposure' (L52) because the authors do not even have data to make predictions on the capacity of phytoplankton on a longer term. Overall, I disagree that the manuscript currently as it is can be published in Nature Communications and suggest the authors to rewrite the manuscript by concentrating solely on the short-term responses at the molecular level on iron and carbon uptake.

We thank the reviewer for their detailed comments and highlighting the insights gained from our short-term molecular data. We agree with the reviewer's assessment that we should refocus more on the short-term molecular-level responses. We have made numerous revisions to our manuscript to emphasize the short-term nature of these experiments, better highlight the short-term molecular-level results, and reduce our conclusions about the long-term effects. For example:

- We have changed our title to highlight that these are short-term responses and removed the word “resistance,” thereby not including any mention of potential long-term effects in the title. It now reads “Short-term acidification promotes diverse iron acquisition and conservation mechanisms in upwelling-associated phytoplankton.”
- We specify in the abstract that we are showing how phytoplankton “respond to short-term acidification exposure” (lines 45-46) and that the mechanisms we describe may confer “short-term resistance” only (line 52). This is also repeated in our discussion on:
 - Lines 546-548: “Within all four experiments however, short-term exposure to pCO₂ levels surpassing global surface seawater means predicted by the end of the century and most natural surface water variability in the CCS (Supplementary Tables 1-3) produced no observable differences”
 - Lines 558-559: “These results may suggest that the natural phytoplankton communities observed here possess short-term resistance to acidification at pH values as low as 7.6 despite iron stress.”
 - Lines 566-567: “However, these results may have occurred from the relatively short duration of the experiment relative to growth rates, particularly at Experiments 2 and 4”
- We again highlight the short-term nature of these experiments at the end of the introduction (lines 151-153) by stating that “these experiments enable probing of mixed phytoplankton assemblages’ sensitivity to short-term acidification exposure.”
- Our discussion has largely been rewritten. Rather than focus on resistance, almost all the first six paragraphs focus on the short-term molecular-level response. Moreover, we directly address the short-term nature of some of our experiments potentially leading to a reduced observable effect on lines 534-536, 566-568, and 572-575.
- We conclude our paper by stating that the tipping points for the effects of OA may exist beyond the concentrations and short time scales used in this experiment on lines 569-575:

“Ocean acidification is in itself a multiple stressor as several carbon and iron parameters in seawater are changing simultaneously (Hurd et al. 2020). As a disturbance, it may be representative of a nonlinear transition where the tipping points are not reached until extreme conditions are encountered (Scheffer et al. 2009). In the case of phytoplankton in upwelling regions, the tipping point may exist at atmospheric CO₂ levels beyond those estimated by the end of the century under high emissions scenarios or more prolonged exposure compared to the short time-scales used here (Supplementary Table 2).”

- We have expanded our analysis of our molecular data by including protein data from additional samples at the second timepoint for Experiment 1. These results are now shown in Figs. 3a, 4a, 4c, and Supplementary Figs. 14, 15, and 22.
- We have added a new Figure 5 that is a cellular schematic highlighting the potential molecular-level changes from acidification-driven changes to iron bioavailability in diatoms.

However, we still believe that it is worthwhile to discuss and make inferences about the longer-term response based on the short-term molecular-level data. These short-term molecular-level changes, particularly since they indicate an iron stress response, offer the ability to make inferences about the longer-term response. We believing following the revisions, we consider multiple potential outcomes and avoid overstating these conclusions. For example:

- It is now clearly stated in the abstract that these short-term results are extrapolated (line 52) and that “we speculate” (line 53) about longer-term exposure resulting in increased iron stress.
- Following our discussion of the short-term molecular-level responses, we discuss potential consequences of using less efficient iron pathways and reducing photosynthetic and nitrogen assimilation proteins as observed in our experiments. Reduced efficiencies for iron uptake, photosynthesis, and nitrogen assimilation would impact growth over longer timescales. However, we further acknowledge that future iron supplies remain unknown; therefore, the potential for increased iron stress remains uncertain. We now state on lines 512-521:

“Collectively, these alternative uptake pathways or iron-sparing measures may have temporarily satisfied the high iron demand of phytoplankton over the relatively short periods examined here, but longer-term exposure to high CO₂ may have resulted in a more iron-stressed community. Siderophore-bound iron or ferrous iron uptake pathways are likely less efficient than those for inorganic ferric iron uptake and a reliance on iron storage is not sustainable over longer time scales. The increase of iron permeases for ferric iron uptake confers increased energetic costs and reduction of iron-requiring photosynthetic and nitrogen assimilation proteins may result in less efficiency in these critical pathways for growth. However, future iron supplies will likely be altered and the net result remains unknown (Boyd et al. 2014; Capone and Hutchins 2013); increased inputs could offset these reductions in iron bioavailability and prevent increased iron stress.”

- We further acknowledge that both iron stress or continued resistance are potential outcomes on lines 567-568:

“longer exposure may have showed either impacts from added iron stress or continued resistance.”

Here, the reviewer also points out several different aspects of our manuscript that indicate a lack of clarity on our part in explaining our experiments and results. In response, we have made several revisions including the addition of new data to resolve these issues with explanations below. We hope that in considering these additions and explanations, the reviewer is confident that iron limitation in phytoplankton was assessed along with ocean acidification within the timescales that they suggest (4 days) at Experiments 3 and 4. As we further detail, we also believe that it is important to consider the community growth rates relative to the duration acidification exposure. Although Experiment 1 was only three days long, the community growth rates were much more rapid suggesting that members of the community performed multiple divisions over the course of the experiment, and there were additional reasons to keep this experiment one day shorter.

1. The reviewer states that the “initial communities of experiment 1 and 2 were iron-limited while the other start communities were not,” and that “experiments 1 and 2 were the ones that only allowed to assess the effect of iron limitation next to OA, but they unfortunately also had the shortest incubation time.” Rather, there were signs of iron stress in all four experiments with a clear signal of iron limitation in Experiments 2, 3, and 4. We apologize that our explanation of the iron status during the time of sampling for our experiments was unclear, and we now include additional data to highlight this:

a. Our new Fig. 1e shows Si_{ex} values throughout the experiments which suggests iron limitation at Experiments 2, 3 and 4. As stated in the manuscript on Lines 183-188:

“ Si_{ex} is an additional biogeochemical proxy that suggests Fe limitation in diatoms due to preferential utilization of silicic acid (Hogle et al. 2018). At Experiment 2, negative values indicating diatom iron stress or limitation were also observed throughout. Following macronutrient addition, negative values were also observed in Experiments 3 and 4 while values remained positive in Experiment 1 (Fig. 1e).”

b. We now show results from a parallel deckboard incubation experiment conducted alongside Experiment 3 (Supplementary Fig. 4d). These results along with the results from the parallel *in situ* incubations for Experiment 2 clearly show the existence of iron limitation in the communities by the first sampling time points (24 hours at Experiment 2 and 48 hours at Experiment 3). As stated on lines 189-193:

“Parallel incubations with unamended control treatments and iron addition treatments (+5 nmol L⁻¹) were performed to further evaluate the presence of iron stress during the experiments. Following 24 hours of incubation at Experiment 2 and 48 hours at Experiment 3, chlorophyll *a* significantly increased with the addition of iron indicating iron limitation in these communities by the first sampling time points (Supplementary Fig. 4).”

- c. We now show results from an additional *in situ* incubation experiment conducted in the vicinity for Experiment 4. Although initiation of this experiment occurred 24 hours following the initiation of the acidification incubations, the seawater was collected while following a Lagrangian drifter making it likely that we used nearly the same water mass and community. As growth was observed after 48 hours, but only following the addition of both iron and nitrate, this experiment shows the existence of iron and nitrogen colimitation. Since nitrate was added to the acidification incubation, the community was no longer nitrogen limited and thus was iron limited throughout the experiment. As stated on lines 193-199:

“At Experiment 4, nitrate was initially depleted indicating no iron limitation in the initial community (Fig. 2 and Supplementary Fig. 5); however, in incubations initiated 24 hours later while following a Lagrangian drifter, significant increases in chlorophyll *a* were observed only with the addition of both iron and nitrate suggesting co-limitation in this community. As nitrate was added without iron in the community subjected to increased acidification, it was likely iron-limited.”

- d. Experiment 1 was not iron-limited as it encountered nitrate depletion after 72 hours; however, we see clear signs of iron stress with diatom molecular indicators (Fig. 4b and Supplementary Fig. 13) In our manuscript, we have refrained from referring to this experiment as iron-limited and directly state that this community encountered macronutrient depletion on lines 199-201, 212-215, 315-318.
 - e. We have added an additional proteomic analysis that examines changes in cellular resource allocations to ribosomal and photosynthetic proteins (Supplementary Fig. 15). These data show that at Experiment 3, centric diatoms and *Ostreococcus* exhibit a significant reduction in photosynthetic protein mass fractions after the initial timepoint which is consistent with iron limitation as observed in diatoms within the Southern Ocean (McCain et al. 2022). In contrast, diatoms reduce their ribosomal protein mass fractions during Experiment 1, likely a result of the onset of nitrogen limitation. We describe these results on lines 314-327 with the methods described on lines 888-898.
2. The reviewer states that there were no observable differences over the short-time scales for Experiment 1 and 2: “this study suffers from insufficient incubation time for exp. 1 and 2, which caused no apparent physiological, transcriptional, proteomic and community differences,” and “the experimental duration of only 2 days did not even allow the phytoplankton cells to adjust their physiological machinery to the applied different environmental conditions (acidification, iron).” We apologize that our results were not clearly explained. In these revisions:
 - a. We hope it is now clear that not only were transcriptional and proteomic changes observed within 48 hours at Experiment 1, but also that this is experiment had

some of the largest changes. These data are shown in Fig. 4a and 4c and Supplementary Table 5. As we now state on Lines 328-330:

“The total number of significantly differentially expressed transcripts was variable across the experiments with the most found at Experiments 1 and 3 where the exposure to acidification relative to the community growth rates were the longest.”

- b. We also hope it is now clear that our iron addition experiments show that these communities were able to make physiological adjustments as observed by increased growth in response to altered iron availability after 24 hours at Experiment 2 and 48 hours for Experiments 3 and 4. These results are described above in reference to the clarification of iron status and are shown in Supplementary Fig. 4. As mentioned above, we likely did not observe a response here at Experiment 1 because it was not iron-limited rather than insufficient incubation time.
3. The reviewer again highlights the short timescale of our incubations as a major point of concern stating “I agree with the authors that the assessment of transcriptomic responses allows already to assess subtle changes within short times (2 days), but then still the question remains whether these responses will be also reflected in phytoplankton growth and biomass build up (chlorophyll, POC). To see these responses at least a duration of 4 days is needed.” and “I agree with the authors that molecular data on iron uptake are interesting and provide interesting insights about the resistance of phytoplankton to OA over a short time scales (between 2 up to max. 4 days), but I disagree that these data will allow to confer the long-term response, as the experimental duration of only 2 days did not even allow the phytoplankton cells to adjust their physiological machinery to the applied different environmental conditions (acidification, iron). Hence, in my opinion, the presented data for experiment 1 and 2 do not even allow to draw any conclusions on the long-term response.”

We again thank the reviewer for highlighting our molecular responses over the relatively short time scales. As described above, we agree that our manuscript should focus more on these molecular-level responses, and have made numerous revisions listed above to do so. We disagree with the reviewer that we should not draw any conclusions on the long-term response, particularly since our observation indicates an iron stress response with potential impacts on iron uptake, photosynthesis, and nitrogen assimilation, but as described above, we have been careful to not overstate our conclusions and consider multiple potential outcomes. There are several additional factors in relation to these two statements that we believe are important to consider here:

- a. Experiments 3 and 4 were four days long thus meeting the minimum requirement suggested by the reviewer and as described above, both experiments exhibited clear signs of Fe limitation.
- b. We disagree that the duration of the experiments should be considered solely with a subjective timescale of 4 days. Rather, we suggest that the duration of exposure to acidification should be considered in the context of community

growth rates. We have now included calculations of community growth rates (μ) estimated from the chlorophyll *a* concentrations as described in the methods on lines 736-739. These results are shown in Supplementary Fig. 7 and indicate that the communities at Experiments 1 and 3 likely were able to perform multiple cell divisions, whereas there was only one doubling in Experiment 4. Experiment 2 was likely too short to have a doubling in the community which was coupled to an initial community that was iron-limited likely leading to a lower growth rate. The results are now mentioned on lines 211-223.

We agree with the reviewer that Experiment 2 was likely too short, but have focused our discussion on the results from the other experiments. Importantly, we believe Experiment 1 is still extremely valuable and showed some of the largest transcriptional and proteomic changes that indicate a stress response that may translate to larger physiological effects. Although this experiment was not iron-limited, there are strong indications of iron stress as observed from the gene expression data shown in Fig. 4b.

- c. There are important advantages in keeping the duration of experiments shorter. We are concerned about bottle effects such as phytoplankton adhering to the walls of the bottle. By lengthening the duration of the experiment, we lose more of the ability to capture the molecular-level changes we are highlighting under conditions that are reasonably reflective of *in situ* conditions. We are also most interested in evaluating phytoplankton responses while they are responding to nutrient availability and growing rather than depleted of macronutrients and encountering stationary growth which ultimately leads to cell senescence. By doing so, we avoid examining responses to OA alongside cell death. Although macronutrient depletion was encountered at the second timepoint at Experiment 1, it was likely early on. Extension of this experiment would have resulted in the conditions we are attempting to avoid.

We hope that in light of these revisions and responses that the reviewer is satisfied that we have largely refocused our manuscript on the short-term molecular-level responses, and that we have assessed ocean acidification along with iron stress within relevant time scales. We largely agree with the reviewer's comments pertaining to other aspects of our manuscript mentioned below. Please see our responses to their additional comments inline and in blue. References are included at the end of this document.

Abstract:

There is large emphasis put on the fact that OA was investigated in iron-stressed phytoplankton of upwelled waters (L44-46), but as laid out by me before, iron-stressed upwelled phytoplankton was mainly found at the sampling sites of experiment 1 and 2. The latter two experiments, however, suffered from a too short incubation time (2 or 3 days), which avoids to obtain at all potential shifts in phytoplankton growth, nutrient uptake and community composition. From these experiments, however, important conclusions are drawn that “phytoplankton growth, nutrient uptake, and community compositions remained unchanged” (L47-48), which indicates their potential resistance to OA. I agree with the authors that ‘molecular-level responses suggest that resistance to acidification-driven changes in iron bioavailability is facilitated by iron uptake pathways that are less hindered by acidification and strategies that reduce cellular iron demand.’”

Therefore, I strongly suggest the authors to rewrite this manuscript with a focus solely on the short-term response at the molecular level on the different iron uptake strategies. Please note that the authors even point out by themselves that for experiment 2 “insufficient incubation time may not have led to community differences” (L216-217) detected with DESeq2.

The reviewer’s concerns listed here are the same as in their general comments, and we have addressed those above. Please refer to those comments.

In relation to our statement highlighted by the reviewer, “insufficient incubation time may not have led to community differences”, the full sentence in the text was specifically referring to the number of differential abundant ASVs in Experiment 2 irrespective of other potential changes. As described above, we acknowledge the relatively short time scale of Experiment 2, and our results largely focus on Experiments 1 and 3. However, as shown in Fig. 2d, there is not a substantial difference in the number of differential abundant ASVs between Experiment 2 and the other experiments, particularly Experiment 1 and 4. As a result, we have removed this sentence as it was misleading. We thank the reviewer for pointing it out.

Introduction:

L93-100: I disagree with the authors that “A relatively small number of field-based studies have examined the effects of ocean acidification on iron stress in phytoplankton”. In fact, there are various studies that have performed Fe-enrichment experiments simulating OA conditions in low Fe regions such as Shi et al. 2010 Science, Feng et al. 2011 Deep Sea Res, Sugie et al. 2013 Biogeosciences, Endo et al. 2013 JEMBE, Yoshimura et al. 2013 J Oceanogr, Yoshimura et al. 2014 Deep-Sea Res, Trimborn et al. 2017 MEPS, Pausch et al. 2022 Frontiers Mar Sci. Moreover, these studies did not add any chelating agent to induce Fe stress, therefore this section needs to be rephrased.

We agree with the reviewer that our statement is inaccurate and thank the reviewer for pointing out these studies examining the interactions between ocean acidification and iron-limitation. We have now rephrased the text to be more factual and detailed about previous studies that have examined ocean acidification and iron stress simultaneously. We now state on lines 97-112:

“Previous field-based studies examining the effects of ocean acidification on phytoplankton communities under iron stress or limitation, namely within the iron-limited subarctic North Pacific and Southern Ocean, have shown conflicting results. Some experiments have shown that diatom communities are negatively impacted with reduced abundances, growth, and photosynthetic efficiencies (Endo et al. 2015; Pausch et al. 2022; Sugie et al. 2013; Trimborn et al. 2017). Other experiments observed no changes in growth, macronutrient uptake, and community composition (Endo et al. 2013; Yoshimura et al. 2014; Yoshimura et al. 2013). One experiment in the Southern Ocean observed no differences in growth or macronutrient uptake in the whole phytoplankton community; however, centric diatoms appeared to be favoured over pennate diatoms indicating variable responses among more specific taxonomic groups that may not be evident if examining the entire community (Feng et al. 2010). In contrast to these other

experiments, Hopkinson et al. (2010) made pH adjustments to a community in the Gulf of Alaska resulting in modest increases to growth and photosynthetic efficiency in conjunction with downregulation of certain photosynthetic proteins. Other ocean acidification studies have been conducted in regions with relatively high ambient iron concentrations, some of which used artificially high levels of strong iron-binding ligands to induce iron stress (Chen et al. 2004; Hoppe et al. 2013; Mausz et al. 2020; Segovia et al. 2017); however, these ligands also increase the solubility of iron leading to increased overall dissolved iron concentrations (Borer et al. 2005). ”

This new paragraph references all but one of the studies listed by the reviewer above. The reference excluded from this paragraph is Shi et al. (2010) as that study used natural seawater in the laboratory with *T. weissflogii* rather than being a fully field-based experiment that uses natural communities although we describe the results from those experiments elsewhere (lines 89-92):

“Further acidification experiments with natural seawater and a single diatom species, *Thalassiosira weissflogii*, did not produce significant differences in iron uptake rates indicating the presence of natural iron-binding ligands that are unaffected by pH (Shi et al. 2010)”

L113-115: The authors point out that “physiological measurements may not be sensitive enough to detect more subtle change, particularly over relatively short time scales”. To me, it is not really clear why a short-term response is more of relevance than a long-term response. In fact, it is much more of interest to better understand how well microorganisms are adapted on a long-term (up to weeks) to upwelling and low Fe conditions than on a short-term (1-4 days). Moreover, there are also physiological adjustments by microorganisms that can immediately respond, as an example the photochemical quantum yield, commonly used as an important indicator of the physiological status of phytoplankton, can already change within 24h. Also, carbon uptake characteristics (CA, CO₂ and HCO₃⁻ uptake) can adjust very quickly. Hence, in my opinion the statement that molecular tools per se are better than physiological measurements is simply not appropriate.

We agree with the reviewer that we should not state that molecular tools are better than physiological measurements of the whole community. Rather, we intend to emphasize that these measurements of the entire community are often used in a way that is incapable of discriminating between changes only occurring within subsets of the community, e.g. specific phytoplankton groups or genera. This is particularly important as it is likely and supported by other studies that there are differing trends among different phytoplankton groups. We also hope to emphasize that molecular-level knowledge of phytoplankton responses is somewhat limited. As a result, we have removed the text mentioned by the reviewer and now state on lines 131-143:

“The aforementioned field-based experiments employed physiological assessments of the whole phytoplankton community that are incapable of discriminating changes only occurring within a specific taxonomic group. At times, taxon-specific pigment analyses

and microscopic cell counts were employed showing a lack of change in abundances for more specific taxonomic groups or species, but not offering a window into taxon-specific physiological adjustments (Endo et al. 2015; Endo et al. 2013; Hopkinson et al. 2010). As previously mentioned, centric and pennate diatoms exhibited differing trends in their relative abundances in an experiment in the Southern Ocean (Feng et al. 2010) suggesting more taxonomically-specific resolution is needed to evaluate responses to ocean acidification. Transcriptomic and proteomic analyses can provide this greater resolution while also providing insight into the short-term molecular-level changes that may precede physiological changes or differences in community structure. Moreover, molecular-level knowledge of phytoplankton responses to acidification is limited to a small number of taxa under laboratory conditions without considering iron bioavailability or using organisms obtained from upwelling areas (Beszteri et al. 2018; Jones et al. 2013; Valenzuela et al. 2018).”

In relation to the reviewers comment about evaluation short-term vs long-term responses, please see our response to the general comments above where the same concerns were listed. The reviewer further states that it would be of more interest to understand adaptation for up to weeks to upwelling, but we believe it is important to consider that upwelling processes largely do not occur on the timescale of multiple weeks. Previous studies indicate that the complete cycle from upwelling initiation to nutrient depletion often only takes 5-7 days (Dugdale and Wilkerson 1989; MacIsaac et al. 1985; Wilkerson et al. 2006). Furthermore, modelling and experimental studies suggested that nutrient depletion can occur within 72 hours regardless of how much nutrient is upwelled (Wilkerson et al. 2006). Therefore, the time scale of incubation on the order of days as used here is more in line with those of upwelling processes.

Discussion:

L367-381: As mentioned above, the observations on growth, community composition and biomass formation in experiments 1 and 2 cannot be considered due to their too short incubation time.

The reviewer stated this concern in their general comments. Please refer to our previous response to those above. Particularly, we would like to emphasize the importance of considering the community growth rate relative to acidification exposure which was highest at Experiment 1. We agree with the reviewer that Experiment 2 was relatively short; however, our analysis largely focuses on Experiments 1 and 3. Furthermore, we directly address the short-term nature of Experiment 2 on lines 534-536:

“A lack of large observable effects in Experiments 2 and 4 may be a result of reduced exposure relative to growth rates compared to Experiments 1 and 3 where multiple cell divisions may have occurred (Supplementary Fig. 7). Corresponding to this reduced exposure relative to growth rate, there were comparatively fewer transcriptional changes in Experiments 2 and 4 as well (Fig. 4a).”

L388-294: Measurements of internal and external carbonic anhydrase activities for various

phytoplankton groups and species exist and it would be good to take them in the discussion into account.

It is difficult to translate our results into a direct connection between internal and external carbonic anhydrases as there is not consistent conservation between family and location (Matsuda et al. 2017). We state this in our manuscript on lines 439-440. However, we agree with the reviewer that added discussion of external carbonic anhydrase activity is beneficial. We now state on lines 523-528:

“Phytoplankton employ a carbon concentrating mechanism (CCM) that includes carbonic anhydrases (CA) to overcome relatively low concentrations of ambient CO₂ (Matsuda et al. 2017). Extracellular CAs (eCA) can alter the carbonate chemistry in the region immediately around the cell (Chrachri et al. 2018) with activities that span orders of magnitude and scale with cell size (Shen and Hopkinson 2015). This function may potentially overcome impacts from acidification such as maintaining elevated [CO₃²⁻] for iron uptake via phytotransferrin.”

L398-420: Different Fe uptake strategies were observed for the communities of experiment 1 and 3, this is somewhat addressed, but not really in detail. This is also due to the fact that the community of experiment 1 was Fe-limited already at the start of the experiment, whereas the community of experiment 3 only became Fe-limited due to the addition of macronutrients. I think this is important to note, but this fact is completely neglected here in the discussion.

We thank the reviewer for pointing out this critical distinction in the observed Fe uptake strategies between Experiments 1 and 3 and strongly agree that our discussion needs to take note of this difference. We have now included text in our results to state this distinction (lines 368-374):

“The increased expression of these transporters is consistent with the expected decrease in bioavailable free inorganic ferric iron that may result from acidification. At Experiment 1, the increased transporter expression implies an increased reliance on siderophore-bound iron or ferrous iron which may increase unlike ferric iron. At Experiment 3, the response was an instead an increase of a ferric iron permease (FTR) that does not have a carbonate-dependence like phytotransferrin. This increase may represent an increase in cell surface transporters to maintain uptake rates as previously suggested in the diatom *T. weissflogii* (Harrison and Morel 1986).”

We also have added text in the discussion section to further evaluate this difference. We agree with the reviewer that one potential reason for this difference among others is the iron status between the communities at the different experiments, but as we now explain in response to the previous comments by the reviewer and with revisions in the manuscript, rather it was Experiment 1 that was not Fe-limited whereas Experiment 3 was. We now state on lines 487-505:

“The difference in iron transporter expression between the two experiments may have occurred for several reasons. First, the iron status of the communities was distinctly

different; the community at Experiment 1 was not iron-limited whereas the community at Experiment 3 was, although in Experiment 1, the $\text{NO}_3:\text{dFe}$ suggested the potential for iron limitation and high expression of ISIP genes indicates some degree of iron stress (Fig. 4b). Meanwhile, Experiment 1 also had the highest initial dFe concentrations (Fig. 1b). Collectively, this may have allowed the community at Experiment 1 to possess relatively high intracellular and/or stored iron to rely on while subjected to potentially added iron stress from acidification. The observed increased expression of ferrous iron transporters that are potentially intracellular may support this increased reliance on intracellular recycling or release of stored iron to sustain growth over these relatively short timescales (Blaby-Haas and Merchant 2012; Lampe et al. 2018). Second, the iron pools between the experiments were likely distinct although they are largely uncharacterized here. Besides higher dissolved iron concentrations, labile particulate iron concentrations were relatively high at Experiment 1, and dissolution of particulate iron may have supported growth (Fig. 1c and Supplementary Fig. 3). Third, the communities were distinct not only in terms of composition but also functional diversity (Figs. 3c). Only 12.2% and 21.5% of orthologous gene clusters in centric and pennate diatoms respectively were found to be shared between the two experiments (Supplementary Fig. 12). Certainly, there is a great deal of functional diversity within diatoms and their responses may not be equivalent.”

Reviewer #2 (Remarks to the Author):

Lampe et al examine the responses of diatoms to coastal upwelled waters that were experimentally acidified by increased CO₂ concentrations. The goal of the study is to see how the phytoplankton community structure and physiology responds. They hypothesize that since coastal upwelling regions experience increased pH variability in comparison to open ocean regions, the microbial communities may be adapted to handle changes in acidity and associated iron stress. The results fit with this hypothesis and there are largely no significant differences in physiological or growth conditions between the three CO₂ conditions (400, 800, 1200 ppm).

The paper is well written and clearly communicates the results. The science is sound, using well established methods in phytoplankton ecophysiology and molecular methods. The methods are adequate to allow replication and verification of the study.

I applaud this paper for being upfront about negative results. For the most part acidification leads to few or no differences in phytoplankton physiology or induce iron limitation due to acidification. This is true even though macronutrients were added to all conditions to try and push the system into iron limitation. My major concern is that it is difficult to comprehend the robustness of their metatranscriptome results across the four different experiments and conditions. The authors state up front that the metatranscriptome results were variable, and chose to present a handful of examples from select organisms within select time points from select treatments in select experiments. Figure 4 shows this best. To determine the robustness of the transcriptional trends presented in the main text, it would be helpful if the authors provided an index of what proportion of treatments, conditions, and experiments a particular gene was found to be significantly differentially transcribed.

We thank the reviewer for their comments on our manuscript. We agree with the reviewer that our transcriptional and proteomic responses are difficult to parse as each experiment's results are different. Genes that were differentially expressed at one point in time were often not differentially expressed in other time points or experiments. To increase the clarity of molecular-level changes, we have added a new main figure, Figure 5, that shows a cellular schematic of potential molecular changes in diatoms from acidification-driven changes to iron bioavailability.

We believe that the differences in results across experiments occurred for several reasons:

1. There are distinct differences in the community structure and functional repertoires among the experiments. We highlight the significant differences in community structure in Fig. 3C. As part of the revision, we have also performed orthologous gene detection with Orthofinder (*Methods* – Lines 818-819). These results show the high degree of functional diversity among the communities in each experiment and are described on lines 270-273 and 501-505:

“Between 23.8% and 48.2% of orthogroups were unique to the detected protein sequences in each experiment’s metatranscriptome assembly highlighting the diverse functional repertoires within each community (Supplementary Fig. 12).”

“The communities were distinct not only in terms of composition but also functional diversity (Figs. 3c). Only 12.2% and 21.5% of orthologous gene clusters in centric and pennate diatoms respectively were found to be shared between the two experiments (Supplementary Fig. 12). Certainly, there is a great deal of functional diversity within diatoms and their responses may not be equivalent.”

2. Each experiment had differences in the duration of exposure to acidification relative to the community growth rates. We have now included calculations of community growth rates (μ) estimated from the chlorophyll *a* concentrations as described in the methods on lines 736-739. These results are shown in Supplementary Fig. 7 and indicate that the communities at Experiments 1 and 3 likely were able to perform multiple cell divisions, whereas there was only one doubling in Experiment 4. Experiment 2 was likely too short to have a doubling in the community which was coupled to an initial community that was iron-limited likely leading to a lower growth rate. The results are now mentioned on lines 211-223. Corresponding to this reduced exposure to acidification, there was also comparatively less transcriptional change at Experiments 2 and 4 (Fig. 4a). As now stated on lines 534-538:

“A lack of large observable effects in Experiments 2 and 4 may be a result of reduced exposure relative to community growth rates compared to Experiments 1 and 3 where multiple cell divisions may have occurred (Supplementary Fig. 7). Corresponding to this reduced exposure relative to growth rate, there were comparatively fewer transcriptional changes in Experiments 2 and 4 as well (Fig. 4a).”

3. Between Experiments 1 and 3 where the most transcriptional changes were observed, there were distinct differences in the iron status of the communities with Experiment 3 being iron-limited whereas Experiment 1 was not. To make this distinction between the different iron statuses clearer, we have revised Fig. 1 to show Si_{ex} values, a biogeochemical proxy for Fe limitation in diatoms, throughout the experiments (Fig. 1e). We also now include additional parallel incubation data showing that the communities Experiments 3 and 4 were likely iron limited after 48 hours. These results are shown in Supplementary Fig. 5 and described on lines 189-201:

“Parallel incubations with unamended control treatments and iron addition treatments (+5 nmol L⁻¹) were performed to further evaluate the presence of iron stress during the experiments. Following 24 hours of incubation at Experiment 2 and 48 hours at Experiment 3, chlorophyll *a* concentrations significantly increased with the addition of iron indicating the presence of iron limitation in these communities by the first sampling time points (Supplementary Fig. 4). At Experiment 4, nitrate was initially depleted indicating no iron limitation in the

initial community (Fig. 2 and Supplementary Fig. 5); however, in incubations initiated 24 hours later while following a Lagrangian drifter, significant increases in chlorophyll *a* were observed only with the addition of both iron and nitrate suggesting co-limitation in this community. As nitrate was added without iron in the community subjected to increased acidification, it was likely iron-limited. Considering low ($< 1 \mu\text{mol L}^{-1}$) nitrate concentrations after 72 hours in Experiment 1 (Fig. 2), the phytoplankton community there was not iron-limited, although it may have encountered some degree of iron stress.”

We also further discuss how this difference may have contributed to the difference in our observations on lines 488-497:

“The iron status of the communities was distinctly different; the community at Experiment 1 was not iron-limited whereas the community at Experiment 3 was, although in Experiment 1, the $\text{NO}_3:\text{dFe}$ suggested the potential for iron limitation and high expression of ISIP genes indicates some degree of iron stress (Fig. 4b). Meanwhile, Experiment 1 also had the highest initial dFe concentrations (Fig. 1b). Collectively, this may have allowed the community at Experiment 1 to possess relatively high intracellular and/or stored iron to rely on while subjected to potentially added iron stress from acidification. The observed increased expression of ferrous iron transporters that are potentially intracellular may support this increased reliance on intracellular recycling or release of stored iron to sustain growth over these relatively short timescales (Blaby-Haas and Merchant 2012; Lampe et al. 2018).”

4. Experiments 1 and 3 also likely had distinct iron pools. We now state on lines 497-500:

“The iron pools between the experiments were likely distinct although they are largely uncharacterized here. Besides higher dissolved iron concentrations, labile particulate iron concentrations were relatively high at Experiment 1, and dissolution of particulate iron may have supported growth (Fig. 1c and Supplementary Fig. 3).”

Despite these differences, we highlight the consistency between Experiments 1 and 3 in terms of using alternative iron uptake pathways that may be less impacted from acidification compared to phytoferritin (McQuaid et al. 2018) and reductions in cellular iron requirements by reducing iron-requiring proteins or using iron-free functional equivalents. This consistency is stated at several points in the manuscript including the Abstract (lines 48-50), Results, and Discussion. For example, on lines 472-473, we now state:

“Within centric and pennate diatoms, these responses were variable, but included changes to myriad of genes related to iron uptake and cellular iron demand (Fig. 5).”

We largely agree with the reviewer's remaining comments, and hope that our revisions and responses have adequately addressed each one. Please see our responses to the additional comments inline and in blue below followed by references at the end of this document.

A few detailed comments:

The proteomics data seems like a side note in the manuscript and under utilized.

We agree with the reviewer that our proteomic data should be more prominently featured. As part of these revisions:

- We now include high-level relative abundances of taxonomic groups from the proteomic results in comparison to the 18S rRNA and mRNA data in our new Fig. 3A.
- We have analyzed additional protein samples to include the second timepoint of Experiment 1. These results are now shown in Figs. 3a, 4a, 4c and Supplementary Figs. 14, 15, and 22.
- We have reanalyzed all our proteomics data to account for certain post-translational modifications which has allowed us to annotate additional proteins (Methods – Lines 870-873). We now discuss additional proteins such as the photosynthetic proteins petB, psbA, petC, and petH; the mitochondrial protein mitoNEET (mNT); the non-iron using superoxide dismutases, Cu-Zn SOD and Ni-SOD; and the iron storage protein, ferritin. For mitoNEET, we also now include an additional phylogenetic tree (Supplementary Fig. 20), providing insight into the evolutionary origins and relatedness of this protein in phytoplankton.
- We performed an additional proteomic analysis that examines cellular resource allocations to ribosomal and photosynthetic proteins. These results on lines 314-327 and Supplementary Fig. 15 show that both centric and pennate diatoms alter their ribosomal protein mass fractions over time at Experiment 1 and that centric diatoms and *Ostreococcus* alter their photosynthetic mass fractions over time at Experiment 3. These results further highlight the differences in physiological states from differing nutrient limitation between the Experiments and can further explain the variability in responses as described above. Additionally, we observed a moderate increase the in pennate diatom ribosomal protein mass fraction at Experiment 1 under high CO₂ indicating an increase in ribosomal investment from increased acidity (lines 318-320).

The proteomics data may still appear like a side note for two reasons:

1. The proteomics data are lower resolution so we are not capturing as many distinct proteins as we are with transcripts. In fact, the proteomics data relies on the metatranscriptome assembly for annotation. As now stated on lines 277-279 and 878-881:

“Often differentially expressed transcripts were not detected as proteins (Supplementary Table 4), likely due to comparatively lower resolution with mass spectrometry, but some consistent patterns were observed.”

“Unique peptides were assigned to a taxonomic and functional annotation based on the metatranscriptome assembly and annotation. Non-unique peptides, i.e., peptides matching to more than one taxonomic group or functional annotation, were considered to be ambiguous.”

2. In comparison to the transcriptome data, we also have fewer samples. Specifically, we do not have proteomic samples from any of our 800 ppm treatments and any samples from Experiments 2 and 4. This is due to lack of access to additional time on a mass spectrometer to analyze these remaining samples, and unfortunately, we are unable to include these samples in our analysis. This is described on lines 274-275 and 282-283:

“In each experiment, differential expression of KEGG-annotated transcripts and proteins was examined between CO₂ treatments within these taxonomic groups. The same was conducted for protein abundances in Experiments 1 and 3...protein abundances were not examined for the 800 ppm treatments.”

Considering that the largest transcriptional changes were observed at Experiments 1 and 3 at 1200 ppm vs 400 ppm CO₂, we believe that focusing on those two experiments and treatments for the proteomic results likely allowed us to quantify more of the molecular effects than with the other experiments, although we acknowledge that there are potentially missed differences from not evaluating the other two experiments. Again unfortunately, we will not be able to include those samples in our analysis, but we hope that the reviewer finds that we have better leveraged the samples added to our analysis at Experiment 1 and the previously existing data.

Figures 1 and 2 were cutoff in the pdf.

We apologize that these two figures were cut off in the merged file. The new figures are no longer cutoff. All figures are also submitted separately and downloadable as individual PDF files separate from the text in their original form.

Figure 3A and B could use some clarification. Is this just all the samples from the four time points combined into a single plot for each experiment? If so, why show the data this way which obscures the temporal trends in community composition.

We agree with the reviewer that these figures from our previous submission are unclear. The reviewer is also correct that the previous Figures 3a and 3b showed the total abundances across all samples within each experiment, and in doing so, there is no indication of any temporal trends. We have removed these figure panels and replaced them with a new panel in Figure 3 (Fig. 3a) showing relative abundances within each time point and treatment across our different data sets.

Ln 196. You mention that SAR86 is the prevalent member of the gamma proteobacteria. For the alphaproteobacteria did you see SAR11 was most abundant? I ask given I was surprised to see that the gammas are so much more abundant than the alphas in fig 3 and I'm wondering if the primer sets chosen are biased against the SAR11 clade members?

Within Alphaproteobacteria, SAR11 was present in all of our samples, but Rhodobacterales were more abundant within each experiment. We now describe the high presence of Rhodobacterales on lines 238-240:

“Alphaproteobacteria and Gammaproteobacteria, were prevalent in all four experiments as observed in the global ocean (Fig. 3a). In particular, these groups were dominated by the ubiquitous *Rhodobacterales* order and SAR86 clade (Supplementary Fig. 9).”

We have also added Supplementary Fig. 9 that shows the relative abundance of different bacterial classes and orders. This figure shows that SAR11 were a relatively more minor constituent of the bacterial communities.

With respect to bias against SAR11, it has previously been shown that the primers used here (515F-Y and 926R) are less biased against SAR11 compared to the popular primers used in the Earth Microbiome Project (515F and 806R)(Parada et al. 2016). The primers used here have also been shown to have the best performance in matching sequences from marine metagenomes among commonly used primers (McNichol et al. 2021). As a result, we believe our primer selection is optimal for this study and do not believe there was any measurable bias against SAR11 in our data.

References

- Beszteri, S., S. Thoms, V. Benes, L. Harms, and S. Trimborn. 2018. The response of three Southern Ocean phytoplankton species to ocean acidification and light availability: a transcriptomic study. *Protist* **169**: 958-975.
- Blaby-Haas, C. E., and S. S. Merchant. 2012. The ins and outs of algal metal transport. *Biochim Biophys Acta Mol Cell Res* **1823**: 1531-1552.
- Borer, P. M., B. Sulzberger, P. Reichard, and S. M. Kraemer. 2005. Effect of siderophores on the light-induced dissolution of colloidal iron (III)(hydr) oxides. *Mar Chem* **93**: 179-193.
- Boyd, P. W., S. T. Lennartz, D. M. Glover, and S. C. Doney. 2014. Biological ramifications of climate-change-mediated oceanic multi-stressors. *Nat Clim Change* **5**: 71.
- Capone, D. G., and D. A. Hutchins. 2013. Microbial biogeochemistry of coastal upwelling regimes in a changing ocean. *Nature Geosci* **6**: 711-717.
- Chen, M., W.-X. Wang, and L. Guo. 2004. Phase partitioning and solubility of iron in natural seawater controlled by dissolved organic matter. *Global Biogeochem Cy* **18**.
- Chrachri, A., B. M. Hopkinson, K. Flynn, C. Brownlee, and G. L. Wheeler. 2018. Dynamic changes in carbonate chemistry in the microenvironment around single marine phytoplankton cells. *Nat Comm* **9**: 74.
- Dugdale, R. C., and F. P. Wilkerson. 1989. New production in the upwelling center at Point Conception, California: temporal and spatial patterns. *Deep-Sea Res* **36**: 985-1007.
- Endo, H., K. Sugie, T. Yoshimura, and K. Suzuki. 2015. Effects of CO₂ and iron availability on *rbcl* gene expression in Bering Sea diatoms. *Biogeosciences* **12**: 2247-2259.
- Endo, H., T. Yoshimura, T. Kataoka, and K. Suzuki. 2013. Effects of CO₂ and iron availability on phytoplankton and eubacterial community compositions in the northwest subarctic Pacific. *J Exp Mar Biol Ecol* **439**: 160-175.
- Feng, Y. and others 2010. Interactive effects of iron, irradiance and CO₂ on Ross Sea phytoplankton. *Deep Sea Res Part I Oceanogr Res Pap* **57**: 368-383.
- Harrison, G. I., and F. M. Morel. 1986. Response of the marine diatom *Thalassiosira weissflogii* to iron stress. *Limnol Oceanogr* **31**: 989-997.
- Hogle, S. L. and others 2018. Pervasive iron limitation at subsurface chlorophyll maxima of the California Current. *Proc Nat Acad Sci USA* **115**: 13300-13305.
- Hopkinson, B. M., Y. Xu, D. Shi, P. J. McGinn, and F. M. M. Morel. 2010. The effect of CO₂ on the photosynthetic physiology of phytoplankton in the Gulf of Alaska. *Limnol Oceanogr* **55**: 2011-2024.
- Hoppe, C. J. M., C. S. Hassler, C. D. Payne, P. D. Tortell, B. Rost, and S. Trimborn. 2013. Iron Limitation Modulates Ocean Acidification Effects on Southern Ocean Phytoplankton Communities. *PLOS ONE* **8**: e79890.
- Hurd, C. L. and others 2020. Ocean acidification as a multiple driver: how interactions between changing seawater carbonate parameters affect marine life. *Mar Freshw Res* **71**: 263-274.
- Jones, B. M. and others 2013. Responses of the *Emiliania huxleyi* Proteome to Ocean Acidification. *PLOS ONE* **8**: e61868.
- Lampe, R. H. and others 2018. Different iron storage strategies among bloom-forming diatoms. *Proc Natl Acad Sci USA* **115**: E12275-E12284.
- MacIsaac, J. J., R. C. Dugdale, R. T. Barber, D. Blasco, and T. T. Packard. 1985. Primary production cycle in an upwelling center. *Deep-Sea Res* **32**: 503-529.
- Matsuda, Y., B. M. Hopkinson, K. Nakajima, C. L. Dupont, and Y. Tsuji. 2017. Mechanisms of carbon dioxide acquisition and CO₂ sensing in marine diatoms: a gateway to carbon metabolism. *Philos Trans R Soc Lond B* **372**: 20160403.
- Mausz, M. A., M. Segovia, A. Larsen, S. A. Berger, J. K. Egge, and G. Pohnert. 2020. High CO₂ concentration and iron availability determine the metabolic inventory in an *Emiliania huxleyi*-dominated phytoplankton community. *Environ Microbiol* **22**: 3863-3882.

- McCain, J. S. P., A. E. Allen, and E. M. Bertrand. 2022. Proteomic traits vary across taxa in a coastal Antarctic phytoplankton bloom. *The ISME Journal* **16**: 569-579.
- McNichol, J., P. M. Berube, S. J. Biller, and J. A. Fuhrman. 2021. Evaluating and Improving SSU rRNA PCR Primer Coverage for Bacteria, Archaea, and Eukaryotes Using Metagenomes from Global Ocean Surveys. *bioRxiv*: 2020.2011.2009.375543.
- McQuaid, J. B. and others 2018. Carbonate-sensitive phytoferritin controls high-affinity iron uptake in diatoms. *Nature* **555**: 534.
- Parada, A. E., D. M. Needham, and J. A. Fuhrman. 2016. Every base matters: assessing small subunit rRNA primers for marine microbiomes with mock communities, time series and global field samples. *Environ Microbiol* **18**: 1403-1414.
- Pausch, F., F. Koch, C. Hassler, A. Bracher, K. Bischof, and S. Trimborn. 2022. Responses of a Natural Phytoplankton Community From the Drake Passage to Two Predicted Climate Change Scenarios. *Frontiers in Marine Science* **9**.
- Scheffer, M. and others 2009. Early-warning signals for critical transitions. *Nature* **461**: 53-59.
- Segovia, M. and others 2017. Iron availability modulates the effects of future CO₂ levels within the marine planktonic food web. *Mar Ecol Prog Ser* **565**: 17-33.
- Shen, C., and B. M. Hopkinson. 2015. Size scaling of extracellular carbonic anhydrase activity in centric marine diatoms. *Journal of Phycology* **51**: 255-263.
- Shi, D., Y. Xu, B. M. Hopkinson, and F. M. M. Morel. 2010. Effect of Ocean Acidification on Iron Availability to Marine Phytoplankton. *Science* **327**: 676-679.
- Sugie, K., H. Endo, K. Suzuki, J. Nishioka, H. Kiyosawa, and T. Yoshimura. 2013. Synergistic effects of pCO₂ and iron availability on nutrient consumption ratio of the Bering Sea phytoplankton community. *Biogeosciences* **10**: 6309-6321.
- Trimborn, S. and others 2017. Iron sources alter the response of Southern Ocean phytoplankton to ocean acidification. *Marine Ecology Progress Series* **578**: 35-50.
- Valenzuela, J. J., A. López García de Lomana, A. Lee, E. V. Armbrust, M. V. Orellana, and N. S. Baliga. 2018. Ocean acidification conditions increase resilience of marine diatoms. *Nat Comm* **9**: 2328.
- Wilkerson, F. P., A. M. Lassiter, R. C. Dugdale, A. Marchi, and V. E. Hogue. 2006. The phytoplankton bloom response to wind events and upwelled nutrients during the CoOP WEST study. *Deep-Sea Res. Part II-Top. Stud. Oceanogr.* **53**: 3023-3048.
- Yoshimura, T., K. Sugie, H. Endo, K. Suzuki, J. Nishioka, and T. Ono. 2014. Organic matter production response to CO₂ increase in open subarctic plankton communities: Comparison of six microcosm experiments under iron-limited and -enriched bloom conditions. *Deep Sea Research Part I: Oceanographic Research Papers* **94**: 1-14.
- Yoshimura, T. and others 2013. Impacts of elevated CO₂ on particulate and dissolved organic matter production: microcosm experiments using iron-deficient plankton communities in open subarctic waters. *Journal of Oceanography* **69**: 601-618.

REVIEWER COMMENTS

Reviewer #3 (Remarks to the Author):

The revised manuscript "Short-term acidification promotes diverse iron acquisition and conservation mechanisms in upwelling-associated phytoplankton" is an interesting set of incubation experiments to address short term acidification on coastal ocean communities. I was not one of the original reviewers, but the revised version generally reads well and it looks like the authors have addressed the comments provided by the past reviewers.

Some specific comments/suggestions based on my readings of this revised version:

- (1) I don't understand why the experiments were done with different durations – it makes cross comparing the results far more complicated. But since all of the experiments did 48 h just comparing those time points would have made for a cleaner story. Related, and I might have missed it but only certain time points were presented in Fig 4C – the supplementary figures should have all of time points/populations. Maybe it's moot now.
- (2) Line 52 – This final sentence doesn't follow from the previous ones – I think there should be some additional information (e.g. that the short term resistance is somehow not sustainable)
- (3) Line 184: The SiEx term needs to be better described in the main text. The sentence where it is first used does not make sense (to me at least). Maybe just add some clarifying wording to the that section.
- (4) Figure 2A – experiment 4 – the significance bar is between 400 and 800?
- (5) Lines 226-244 and figure 3- This section of text and the associated figure is conflating RNA based activity with DNA based abundance. Sure if an RNA is present it means that the organism is there, but RNA transcript numbers (either rRNA or mRNA) cannot be used as a metric of community composition per se. For example, in experiment 4 (line 244), with this data we don't know if there is a higher presence of cyanobacteria or if they were just more 'active.' This section and figure needs to be revised to remove references to abundance and community composition and to focus this solely on 'activity' by community members.

Overall this is a solid contribution.

Reviewer #4 (Remarks to the Author):

22-7-2023

Review for "Short-term acidification promotes diverse iron acquisition and conservation mechanisms in upwelling-associated phytoplankton"

Stepping in late in the review of the wonderful study by Lampe and many talented and invested collaborators, I will take a bit unorthodox approach to the review. Rather than outlining many specific points, I'd like to review the paper & study strength and limitations, and allow the authors and editors take their choices.

I will first briefly outline the main points and then elaborate on a few. I see these as philosophical & strategical and leave it for the authors and editors to find the best path forward.

The main strengths of the paper are:

- A. Carefully conducted and successful set of complicated experiments in natural waters, combining many advanced and high-quality analyses (chemical parameters, physiological, OMICS and more)
- B. Focus on Fe transport, storage and use by natural eukaryotes, combining transcript level and uptake rates by highly knowledgeable researchers.
- C. Four experiments varying in community composition and physiological state, which yielded different set of responses to the nutrient addition/incubation and to increasing CO₂, thus better informing us on the range of natural responses.
- D. Brave choice to be upfront about negative results (important!!). As a whole the paper is very honest and does a great job keeping high level of complexity, while still conveying clear messages.

The main weaknesses are:

- A. The authors are still spending too much effort and space on the link between ocean acidification and iron availability / uptake. The last sentence in the abstract is indeed a long-shot... The abstract as a whole is focused only on the effect of acidification, while in fact I think this paper has more to offer (see next)
- B. Complexity – many experiments, many variables, many techniques. It is clear that huge efforts were made to simplify it all and streamline it, but...
- C. Synthesis between uptake rates and upregulation of uptake transporters (between experiments, throughout the experiment) can be improved (see some ideas next)

Below I expand on two issues:

1. Better integration of uptake rates and Fe gene regulation
This is clearly one of the major goals of the paper and indeed the authors quite adequately discuss it (well done!). I did not find much on rate comparison between the experiments (and perhaps throughout the experiments). This is partly as the paper presents uptake rates normalized to POC and not to Chl, which are more homogenous. I did not find an explanation, but since we are dealing with phytoplankton, Chl may be more appropriate (at least for comparison among experiments). These rates are more variable and perhaps offer some insights.

Below, I present a quick analysis, which is just a suggestion and a possible direction to follow. I ignored acidification and compared ^{59}Fe uptake rates per Chl among experiments. Starting with FeCl_3 (higher uptake rates & hence more reliable, and also more likely to represent larger/more dynamic Fe pools that may better match transcripts of several transporters). In this analysis, my assumption is that short term uptake rate can report on the Fe limitation status of the cells (i.e. higher rates indicate that the cells are more stressed and hence upregulate more transporters). If this is the case, we expect to see a link between measured rate and transporters expression—either between exp, or during time in each exp. Ranking FeCl_3 uptake/chl rates among exp (not including To) we find that $E_4 > E_3 > E_2 > E_1$. This would indicate (as I think stated in the paper based on other data) that cells in E1&2 are less limited than E3&4. However, when FeCl_3 uptake rates are normalized to POC the data changes dramatically with $E_1 > E_3 = E_4 > E_2$. (I vote for chl) Looking at trend with time E3 and E4 are stable, while uptake rates drop in E1, E2. This does not go very well with the presumed onset of Fe limitation in E1, but.. Then FeDFB uptake rates, much discussion but no mention that in E2 you had uptake rates (normalized for both C & Chl) which are an order of magnitude slower than any other exp. Can you find any agreement with transcript patterns / changes in species composition etc?

2. Emphasizing other highlights/achievements of the study – I think that we are all very data driven, but in fact new scientific approaches, and particularly the combination of knowledge across techniques, scales, functions etc, are good enough reason to get the publication to Nature Communications (or other high impact). I think we are far from resolving the effect of acidification on iron bioavailability to diverse phytoplankton in natural systems, due to gaps in our knowledge on Fe chemistry (what are the major forms of Fe out there?), phytoplankton types (who is there and how active), coping mechanisms and more. Every sensible reader (or editor) should not be surprised that the effect is variable and even contradictory (in general in papers so far). Can you try and refine / extract methodological/conceptual goals achieved (even for one case) which are related to the general question of iron stress / storage/ uptake/ availability? Maybe you can strengthen the contrast between E1 and E3, emphasizing the differences in status, composition and response, but emphasize a bit more the set of tools / knowledge gained that allow you to “finger-print” the Fe status and response of this community during the incubation.

Good luck and congratulations on this important scientific contribution.

Reviewer #3 (Remarks to the Author):

The revised manuscript "Short-term acidification promotes diverse iron acquisition and conservation mechanisms in upwelling-associated phytoplankton" is an interesting set of incubation experiments to address short term acidification on coastal ocean communities. I was not one of the original reviewers, but the revised version generally reads well and it looks like the authors have addressed the comments provided by the past reviewers.

Some specific comments/suggestions based on my readings of this revised version:

We thank the reviewer for their time and comments on our manuscript. Please see our responses inline and in blue followed by references at the end of this document.

(1) I don't understand why the experiments were done with different durations – it makes cross comparing the results far more complicated. But since all of the experiments did 48 h just comparing those time points would have made for a cleaner story. Related, and I might have missed it but only certain time points were presented in Fig 4C – the supplementary figures should have all of time points/populations. Maybe it's moot now.

The experiments were done with different durations to attempt to avoid macronutrient depletion as we are most interested in evaluating phytoplankton while they are responding to nutrient availability and growing rather than encountering stationary growth which ultimately leads to cell senescence. By doing so, we avoid examining responses to acidification alongside cell death. Extension of Experiment 1 would have resulted in the conditions we were attempting to avoid. To clarify these reasons, we now state on lines 169-172:

“Samples were then collected at one-day intervals that varied among experiments to attempt to avoid macronutrient depletion with an upper limit of 96 hours to also avoid increased bottle effects: 48 and 72 hours for experiment 1, 24 and 48 hours for experiment 2, and 48 and 96 hours for experiments 3 and 4 (Table 1).”

We also now explain these reasons in the methods section on lines 654-658:

“For the later time points, sampling occurred at one-day intervals that varied among experiments: 48 and 72 hours for Experiment 1, 24 and 48 hours for Experiment 2, and 48 and 96 hours for Experiments 3 and 4 (Table 1). These intervals were chosen to attempt to avoid macronutrient depletion with a maximum experimental duration of 96 hours to also avoid increased bottle effects.”

Although the time points vary, direct comparisons where time points overlap is still conflated by several factors. The initial environmental conditions were largely different across the wide range of variables that were assessed (Figs. 1 and 2 and Supplementary Figs. 1-3). We also examined distinct microbial communities not only in terms of composition (16S and 18S rRNA, Fig. 3c) but also in terms of functional diversity (Supplementary Fig. 13). Collectively, these differences were likely also responsible for the observed differing community growth rates (Supplementary Fig. 7); thus, the exposure to acidification relative

to growth rate varied among the experiments as well (discussed on lines 228-239, 345-347, and 572-574).

Rather than focus on the 48-hour time points, which would largely exclude time points where there is a longer duration of exposure to acidification, we have opted to showcase samples where we have observed interesting changes in differential expression, and are omitting samples where differential expression was relatively low as shown in Fig. 4a or are largely of unknown function (lines 351-352). We describe the plots in Figure 4c in the caption as such:

“MA plots for taxonomic groups, experiments (E), and time points (T), where significant differential expression for transcripts (top row) or proteins (bottom row) of interest between the 1200 and 400 ppm treatments was observed.”

However, we certainly agree with the reviewer that we should be presenting more of our data, particularly where there is differential expression and there is paired transcript and protein data. In addition to the data presented in Fig. 4, the previous version of our manuscript included Supplementary Figures showing:

- Transcript expression of iron-starvation induced proteins 1, 2A, and 3 in pennate diatoms (now Supplementary Figs. 14) as well as corresponding protein expression in both centric and pennate diatoms (now Supplementary Fig. 15)
- Differential expression of nitrogen assimilation transcripts for pennate diatoms at Experiments 1 and 3, centric diatoms and Experiment 3, and *Ostreococcus* at Experiment 3 (now Supplementary Fig. 25)
- Transcript and protein expression for *Ostreococcus* at the second time point (T96) of Experiment 3 (now Supplementary Fig. 26)
- SLC4-family bicarbonate transporter expression in centric and pennate diatoms (now Supplementary Fig. 27)

We have now added new figures showing:

- Differential transcript and protein expression for centric diatoms for the time points of Experiments 1 and 3 not shown in Fig. 4c. as Supplementary Fig. 23
- Ferrichrome-binding protein 1 (FBP1) expression from all diatoms for both transcripts (Supplementary Fig. 21) and proteins (Supplementary Fig. 22). These data are also now discussed in the manuscript on lines 400-403 and 512-515 in response to comments by Reviewer 4 (see below).

(2) Line 52 – This final sentence doesn’t follow from the previous ones – I think there should be some additional information (e.g. that the short term resistance is somehow not sustainable)

We agree and thank the reviewer for pointing out this omission. We have expanded this sentence to now state:

“By extrapolating these short-term results, we speculate that these mechanisms may only temporarily satisfy cellular iron demand, and longer-term exposure to acidification

without increased iron inputs may result in increased iron stress for phytoplankton communities.”

(3) Line 184: The Si_{ex} term needs to be better described in the main text. The sentence where it is first used does not make sense (to me at least). Maybe just add some clarifying wording to the that section.

We agree that this text was unclear and have expanded it to better explain the Si_{ex} proxy. We now state on lines 186-191:

“Si_{ex} is an additional biogeochemical proxy that serves as an indicator of Fe limitation in diatoms; as Fe-limited diatoms preferentially uptake silicic acid (H₄SiO₄) relative to NO₃, lower than expected H₄SiO₄ concentrations compared to NO₃ concentrations indicates the presence of Fe limitation. Si_{ex} reports the difference between H₄SiO₄ and NO₃ concentrations (Methods); therefore, negative values indicate greater H₄SiO₄ uptake due to iron stress in diatoms.”

(4) Figure 2A – experiment 4 – the significance bar is between 400 and 800?

Upon revisiting these data, we realized that the significant difference in Figure 2A was reported in error, and that the means are not significantly different. We have also revisited all the data and statistical tests in Figure 2 to ensure that the remaining information is accurate. We thank the reviewer for helping us find this error and have updated Figure 2A to remove the indication of significance there. This change does not affect any of the text in the manuscript; therefore, no additional revisions in relation to this comment have been made.

(5) Lines 226-244 and figure 3- This section of text and the associated figure is conflating RNA based activity with DNA based abundance. Sure if an RNA is present it means that the organism is there, but RNA transcript numbers (either rRNA or mRNA) cannot be used as a metric of community composition per se. For example, in experiment 4 (line 244), with this data we don't know if there is a higher presence of cyanobacteria or if they were just more 'active.' This section and figure needs to be revised to remove references to abundance and community composition and to focus this solely on 'activity' by community members.

We respectfully disagree with the reviewer that we cannot use rRNA abundance to assess community composition and further disagree that we should focus on activity for the following reasons:

1. The use of RNA instead of DNA attempts to avoid measuring abundance from deceased cells (Blazewicz et al. 2013), and therefore, may provide a better assessment of microbial community diversity (Hu et al. 2016; Kong et al. 2023).
2. DNA-based assessments are biased by high copy number variation (Gong and Marchetti 2019; Větrovský and Baldrian 2013), although as the reviewer notes, we agree that RNA-based abundances are affected by variable rRNA expression. In essence, both approaches have a similar but different underlying bias that creates a disconnect between rDNA or rRNA and true cellular abundance, although it remains

- possible that rRNA is less biased in this regard. As rRNA is a structural component of the cell's ribosomes, it remains a potentially useful proxy for biomass.
3. It has been suggested that rRNA is not a reliable indicator of activity (Blazewicz et al. 2013). There is variability between rRNA concentration and growth rates, and for some organisms, there is no relationship at all. Dormant cells may also possess high numbers of ribosomes and therefore potentially high rRNA copies (Blazewicz et al. 2013).
 4. Although less common than rDNA, likely due to added costs or complexity to an extent, other previously published papers have used RNA-based metabarcoding to examine community composition, and at times, in conjunction with DNA-based metabarcoding. For example:
 - Cohen, N. R., M. R. McIlvin, D. M. Moran, N. A. Held, J. K. Saunders, N. J. Hawco, M. Brosnahan, G. R. DiTullio, C. Lamborg, J. P. McCrow, C. L. Dupont, A. E. Allen and M. A. Saito (2021). "Dinoflagellates alter their carbon and nutrient metabolic strategies across environmental gradients in the central Pacific Ocean." *Nature Microbiology* 6(2): 173-186.
 - Traving, S. J., O. Rowe, N. M. Jakobsen, H. Sørensen, J. Dinasquet, C. A. Stedmon, A. Andersson and L. Riemann (2017). "The Effect of Increased Loads of Dissolved Organic Matter on Estuarine Microbial Community Composition and Function." *Frontiers in Microbiology* 8: 351.
 - Hu, S. K., V. Campbell, P. Connell, A. G. Gellene, Z. Liu, R. Terrado and D. A. Caron (2016). "Protistan diversity and activity inferred from RNA and DNA at a coastal ocean site in the eastern North Pacific." *FEMS Microbiology Ecology* 92(4): fiw050.
 - Needham, D. M., E. B. Fichot, E. Wang, L. Berdjeb, J. A. Cram, C. G. Fichot and J. A. Fuhrman (2018). "Dynamics and interactions of highly resolved marine plankton via automated high-frequency sampling." *The ISME Journal* 12(10): 2417-2432.

We are not advocating for rRNA as a more suitable alternative to rDNA. Rather, we are suggesting that both approaches offer a molecular-level assessment of microbial community composition, each with their own caveats. Furthermore, we agree with the reviewer it should be clear to the reader that an rRNA-based approach was used so that it is not conflated with DNA-based abundance. We now:

- State in the results that “community compositions...were inferred via 18S and 16S ribosomal RNA (rRNA) amplicon sequencing” (lines 242-243), and further specify that we are examining “rRNA” and “rRNA-derived amplicon sequence variants.” (lines 244 and 271 and Fig. 3 caption).
- State in the methods text (lines 817-820): “From the aforementioned extracted RNA, cDNA was synthesized with the Invitrogen SuperScript™ III First-Strand Synthesis System according to the manufacturer’s instructions. RNA-derived amplicon libraries targeting the V4-V5 hypervariable region of the 16S gene and the V9 hypervariable region of the 18S region were then generated...”

Lastly, with respect to the reviewer's comment that we are not sure if there were more cyanobacteria present or if they were just active, there is unpublished flow cytometry data throughout the same cruise that supports our assessment of the increased abundance of cyanobacteria in the vicinity of Experiments 3 and 4, but we are unable to include these data in our manuscript.

Overall this is a solid contribution.

We again thank the reviewer for their time and constructive feedback.

Reviewer #4 (Remarks to the Author):

Stepping in late in the review of the wonderful study by Lampe and many talented and invested collaborators, I will take a bit unorthodox approach to the review. Rather than outlining many specific points, I'd like to review the paper & study strength and limitations, and allow the authors and editors take their choices.

I will first briefly outline the main points and then elaborate on a few. I see these as philosophical & strategic and leave it for the authors and editors to find the best path forward.

The main strengths of the paper are:

- A. Carefully conducted and successful set of complicated experiments in natural waters, combining many advanced and high-quality analyses (chemical parameters, physiological, OMICS and more)
- B. Focus on Fe transport, storage and use by natural eukaryotes, combining transcript level and uptake rates by highly knowledgeable researchers.
- C. Four experiments varying in community composition and physiological state, which yielded different set of responses to the nutrient addition/incubation and to increasing CO₂, thus better informing us on the range of natural responses.
- D. Brave choice to be upfront about negative results (important!!). As a whole the paper is very honest and does a great job keeping high level of complexity, while still conveying clear messages.

We thank the reviewer for their thoughtful comments. Please see our responses inline in blue below followed by references at the end.

The main weaknesses are:

- A. The authors are still spending too much effort and space on the link between ocean acidification and iron availability / uptake. The last sentence in the abstract is indeed a long-shot... The abstract as a whole is focused only on the effect of acidification, while in fact I think this paper has more to offer (see next)

We thank the reviewer for pointing out that the study has more to offer and agree. As the reviewer suggests and we describe in greater detail below (see comment 3), we have further highlighted the methodologies used here to assess the phytoplankton communities' responses and strengthened our analysis and discussion to better integrate the Fe uptake rates and transcript/protein abundances. We also further highlight our phylogenetic characterization of several relatively uncharacterized iron-related genes in diatoms that were observed to be part of the acidification-driven response here.

However, we respectfully disagree with the reviewer that we are spending too much time on the connections between ocean acidification and iron bioavailability. As described in our introduction, there is enormous uncertainty surrounding phytoplankton responses to ocean acidification, particularly when considering relationships between ocean acidification and iron bioavailability (lines 71-145). These experiments were designed to specifically address this topic and detected an iron-related response indicative of acidification-driven changes to iron bioavailability. Thus, we prefer to keep the manuscript focused on these connections and

associated responses, although we largely agree with the comments detailed below and have made several revisions to address them.

With respect to the comment about the last sentence of the abstract, we agree that it is speculative as stated within the sentence. In response to a comment by Reviewer 3, we also have modified this sentence to increase clarity. It now reads:

“By extrapolating these short-term results, we speculate that these mechanisms may only temporarily satisfy cellular iron demand, and longer-term exposure to acidification without increased iron inputs may result in increased iron stress for phytoplankton communities.”

We discuss this topic in more detail in our discussion on lines 547-557 where we suggest that the mechanisms employed in response to increased acidification confer less efficiency for iron uptake, photosynthesis, and nitrogen assimilation or greater energetic costs while acknowledging that future iron supplies remain unknown where decreased iron bioavailability could be offset by increased iron inputs. We further acknowledge that it remains possible that continued resistance could be observed over longer time scales (lines 603-605). We believe it is worthwhile to discuss and make inferences about the longer-term responses based on these short-term molecular data, particularly as the molecular-level responses may precede an observable physiological response or shift in community structure, and our text considers multiple potential outcomes to avoid overstating these conclusions.

B. Complexity – many experiments, many variables, many techniques. It is clear that huge efforts were made to simplify it all and streamline it, but...

C. Synthesis between uptake rates and upregulation of uptake transporters (between experiments, throughout the experiment) can be improved (see some ideas next)

As the reviewer expands on these comments below, we have addressed those separately.

Below I expand on two issues:

1. Better integration of uptake rates and Fe gene regulation

This is clearly one of the major goals of the paper and indeed the authors quite adequately discuss it (well done!). I did not find much on rate comparison between the experiments (and perhaps throughout the experiments). This is partly as the paper presents uptake rates normalized to POC and not to Chl, which are more homogenous. I did not find an explanation, but since we are dealing with phytoplankton, Chl may be more appropriate (at least for comparison among experiments). These rates are more variable and perhaps offer some insights.

Below, I present a quick analysis, which is just a suggestion and a possible direction to follow. I ignored acidification and compared ^{59}Fe uptake rates per Chl among experiments. Starting with FeCl_3 (higher uptake rates & hence more reliable, and also more likely to represent larger/more dynamic Fe pools that may better match transcripts of several transporters). In this analysis, my assumption is that short term uptake rate can report on the Fe limitation status of the cells (i.e. higher rates indicate that the cells are more stressed and hence upregulate more transporters). If this is the case, we expect to see a link between

measured rate and transporters expression— either between exp, or during time in each exp. Ranking FeCl₃ uptake/chl rates among exp (not including To) we find that E₄>E₃>E₂>E₁. This would indicate (as I think stated in the paper based on other data) that cells in E₁&E₂ are less limited than E₃&E₄. However, when FeCl₃ uptake rates are normalized to POC the data changes dramatically with E₁>E₃=E₄>E₂. (I vote for chl)

Looking at trend with time E₃ and E₄ are stable, while uptake rates drop in E₁, E₂. This does not go very well with the presumed onset of Fe limitation in E₁, but..

Then FeDFB uptake rates, much discussion but no mention that in E₂ you had uptake rates (normalized for both C & Chl) which are an order of magnitude slower than any other exp. Can you find any agreement with transcript patterns / changes in species composition etc?

The reviewer raises several interesting points within this comment that although related, we summarize and address separately for clarity:

1. Normalization of Fe uptakes to chlorophyll *a* rather than particulate organic carbon (POC)

We agree with the reviewer that it may be more appropriate to prioritize reporting of the chlorophyll-normalized Fe uptake rates over the POC-normalized rates. As a result, we have updated Figure 2d to show the chlorophyll-normalized data rather than the POC-normalized data. Our initial concern with the chlorophyll-normalized data is chlorophyll content variability that is influenced by light, temperature, and nutritional status (expressed as C:Chl)(Wang et al. 2009), although light and temperature are consistent across the experiments when omitting the initial conditions (T₀). As a result, we agree that chlorophyll *a* is more suitable. Furthermore, we are concerned with contributions from heterotrophic biomass to POC although 1 μm filters were consistently used to avoid measuring free-living bacteria. As the initial conditions are likely confounded by chlorophyll variability, we believe it remains worthwhile to include the POC-normalized data as Supplementary Fig. 6 and in Supplementary Fig 7C-D, but as previously mentioned, we now prioritize showing and discussing the chlorophyll-normalized data.

The reviewer is also correct that we did not explain our reasoning in the previous version of the manuscript. To address this, we now state on lines 215-219:

“Although chlorophyll content varies in response to environmental conditions (Wang et al. 2009), light and temperature were consistent during the experiments resulting in comparable chlorophyll-normalized rates that may better reflect photoautotrophic biomass compared to POC. Chlorophyll *a* has often also been used to normalize Fe uptake rates in phytoplankton communities as POC includes heterotrophic biomass (Hoppe et al. 2013; Marchetti and Maldonado 2016; Trimborn et al. 2015).”

Relatedly, the reviewer points out that our Fe-DFB uptake rates are much slower at Experiment 2; however, this observation is only for the POC-normalized data and not the chlorophyll-normalized data (Fig. 2d and Supplementary Fig. 6). This difference is driven by much higher POC concentrations at Experiment 2. As we are now prioritizing the chlorophyll-normalized data that do not show this pattern, and the POC-normalized data

is influenced by heterotrophic biomass, we have chosen to not further explore these results in the manuscript.

2. Comparison of Fe uptakes among and throughout the experiments to report on Fe status

We agree with the reviewer that the Fe uptake rates should also be presented to make comparisons more easily over time and among experiments. As mentioned in comment 1, we have now added Supplementary Fig. 7 to show both the chlorophyll-normalized rates and the POC-normalized rates over time and across experiments with equivalent y-axes.

We also thank the reviewer for making the extremely important suggestion that we better use our Fe uptake rates to assess the Fe nutritional status of the phytoplankton communities. We agree with the reviewer that the maximum short-term Fe uptake rates provides an indication of the number of transporters and thus, iron stress as iron-stressed cells likely upregulate more transporters. The reviewer is also correct that when using the chlorophyll-normalized data and final time points of each experiment, there is an order in FeCl₃ uptake rates where E4>E3>E2>E1. This trend is the same for the chlorophyll-normalized Fe-DFB uptake rates as well. However as the reviewer also notes, we also observed significant reductions in the FeCl₃ uptake rates over time at Experiments 1 and 2. Alone, these uptake rates may suggest that Experiment 2 is less iron-stressed than Experiments 3 and 4; however, our other biogeochemical, experimental, and molecular data are contradictory. Experiment 2 had the high NO₃:dFe ratios and lowest Si_{ex} values (Fig. 1c and 1e). We also experimentally show significant increases in chlorophyll *a* upon the addition of Fe after 24 hours (Supplementary Fig. 4b) and high expression of diatom iron stress marker genes (Fig. 4c and Supplementary Fig. 14).

As we have numerous other lines of evidence including experimental iron additions showing iron limitation in this experiment, we would like to refrain from using the FeCl₃ rates to infer iron status. However, we recognize the value in highlighting that Fe-DFB uptake is also indicative of Fe stress at Experiments 3 and 4 as phytoplankton are believed to only take up significant amounts of Fe-DFB when iron-limited. We now state on lines 224-227:

“The measurable uptake of Fe-DFB, particularly at Experiments 3 and 4 where rates were similar to those of iron-limited populations and isolates (Maldonado and Price 1999; Maldonado and Price 2001), further indicates iron stress as phytoplankton normally only acquire significant amounts of Fe-DFB when iron-limited (Marchetti and Maldonado 2016).”

3. Connections between uptake rates and ‘omic data

We agree with the reviewer that we should show stronger connections between the Fe uptake rates and transcript/protein expression patterns. By now showing the chlorophyll-normalized Fe uptake (see comment 1), we now also describe the significantly higher Fe uptake rates at Experiment 3 between the 1200 and 400 ppm treatments (Fig. 2d). This increase in Fe uptake that indicates an increase in the number of transporters for free inorganic ferric iron is coupled to an observed increase in both iron permease (FTR1)

transcripts and proteins in centric diatoms (Fig. 4c). We now discuss these results at several points throughout the manuscript:

- Lines 220-222: “When normalized to chlorophyll *a*, maximum inorganic Fe uptake rates were significantly higher in the 1200 ppm when compared to the 400 ppm treatment at the final time point for Experiment 3 ($P = 0.016$) indicating an increase in cell surface transporters (Marchetti and Maldonado 2016).”
- Lines 390-394: “At Experiment 3, the response was instead an increase of a ferric iron permease (*FTR1*) that does not have a carbonate-dependence like phytotransferrin. Observed as both transcripts and proteins, this increase suggests that there were a greater number of cell surface transporters for inorganic ferric iron as supported by higher maximum uptake rates of inorganic iron (Fig. 2d).”
- Lines 517-521: “In Experiment 3, the response was instead transcriptional and proteomic increases in iron permeases (*FTR*) in centric diatoms signifying an increase in free inorganic iron transporters as its availability is reduced (Fig. 4c). This increase in transporters is further supported by higher maximal uptake rates of inorganic iron that are also a further sign of increased iron stress (Fig. 2d).”

We have also further analyzed our data to describe and show ferrichrome-binding protein 1 (*FBP1*) expression as it has been directly linked to siderophore uptake including that of Fe-DFB (Supplementary Figs. 21 and 22)(Coale et al. 2019). The lack in change in expression is connected to the similar Fe-DFB uptake rates among treatments:

- Lines 400-403: Ferrichrome-binding protein 1 (*FBP1*) acts as a receptor in concert with ferric reductases for siderophore uptake including Fe-DFB, but it was not differentially expressed (Supplementary Fig. 21 and 22) corresponding to the similar Fe-DFB uptake rates among treatments (Fig. 2d and Supplementary Fig. 6).
- Lines 512-515: Genes responsible for Fe-DFB uptake, *FRE* and *FBP1*, were also not differentially expressed except for decreased protein abundances in centric diatoms in Experiment 1, although the ferric reductases expressed here may represent separate homologs that does not serve the same functional role (Fig. 4c and Supplementary Figs. 21 and 22)(Coale et al. 2019).

Otherwise, the data do not show always show strong connections between expression and uptake rates over time within each experiment. For example, *ISIP2A* expression increases and *FTR* expression is stable while FeCl_3 uptake rates decline at Experiments 1 and 2 (Fig 4b and Supplementary Figs. 14 and 15). We do not observe a strong correlation between *FBP1* expression and Fe-DFB uptake over time either. As a result, we have not made further revisions in relation to this comment.

2. Emphasizing other highlights/achievements of the study – I think that we are all very data driven, but in fact new scientific approaches, and particularly the combination of knowledge across techniques, scales, functions etc, are good enough reason to get the

publication to Nature Communications (or other high impact). I think we are far from resolving the effect of acidification on iron bioavailability to diverse phytoplankton in natural systems, due to gaps in our knowledge on Fe chemistry (what are the major forms of Fe out there?), phytoplankton types (who is there and how active), coping mechanisms and more. Every sensible reader (or editor) should not be surprised that the effect is variable and even contradictory (in general in papers so far). Can you try and refine / extract methodological/conceptual goals achieved (even for one case) which are related to the general question of iron stress / storage/ uptake/ availability? Maybe you can strengthen the contrast between E1 and E3, emphasizing the differences in status, composition and response, but emphasize a bit more the set of tools / knowledge gained that allow you to “finger-print” the Fe status and response of this community during the incubation.

We agree that the text should more clearly describe the methodology and how that enabled us to assess the communities’ iron status and responses. We now:

- State in our abstract that “A combined physiological and multi-omics approach was applied...” (line 47).
- Briefly highlight our approach at the end of our introduction (lines 153-156): "With a combined physiological and multi-omic approach, these experiments enable probing of mixed phytoplankton assemblages’ sensitivity to short-term acidification exposure from the whole community to taxonomically-specific molecular levels while considering the high importance of iron bioavailability.”
- State in the discussion how the combined approaches allowed us to examine the community and identify these responses (lines 496-502): “Here we examined upwelling-associated phytoplankton communities’ responses while they were simultaneously exposed to iron stress and acidification over short time periods. The iron status and responses of these communities were assessed with a combination of experimental, biogeochemical, physiological, and ‘omics approaches. Although large differences were not observed in the bulk community, molecular-level responses with greater taxonomic resolution of dominant groups were indicative of acidification-driven alterations to iron bioavailability and greater iron stress under higher CO₂.”

We also agree that the contrast between Experiments 1 and 3 is interesting and go into further detail in our discussion about the differences in gene expression (lines 507-521 and 550-554) that may be explained by the differences in iron status (lines 523-532) and community composition (lines 535-538) while also highlighting similarities in the observed molecular-level responses (lines 540-547).

In terms of additional goals achieved, we also now further highlight our detection and phylogenetic analysis of several relatively uncharacterized iron-related genes in diatoms that are elucidated to be of diverse evolutionary origins and part of the acidification-driven responses we observed here. We now:

- State in our abstract (lines 50-52): “Underlying these responses are genes of diverse evolutionary origins highlighting unique adaptations in phytoplankton to grow under low iron availability.

- Describe the evolutionary origins of the ZIP-family protein detected here (lines 364-365): “Diatom homologs of the *ZIP1* gene are related to ZIP-family genes in green algae...(Lampe et al. 2018).”
- Emphasize this finding in our discussion (lines 504-506): “Several of the iron-related genes described here are relatively uncharacterized in diatoms and of diverse evolutionary origins highlighting diatoms’ unique adaptations to grow under low iron availability.”

Good luck and congratulations on this important scientific contribution.

We again thank the reviewer for their time and helpful comments.

References

- Blazewicz, S. J., R. L. Barnard, R. A. Daly, and M. K. Firestone. 2013. Evaluating rRNA as an indicator of microbial activity in environmental communities: limitations and uses. *The ISME Journal* **7**: 2061-2068.
- Coale, T. H., M. Moosburner, A. Horák, M. Oborník, K. A. Barbeau, and A. E. Allen. 2019. Reduction-dependent siderophore assimilation in a model pennate diatom. *Proc Natl Acad Sci USA* **116**: 23609-23617.
- Gong, W., and A. Marchetti. 2019. Estimation of 18S Gene Copy Number in Marine Eukaryotic Plankton Using a Next-Generation Sequencing Approach. *Frontiers in Marine Science* **6**.
- Hoppe, C. J. M., C. S. Hassler, C. D. Payne, P. D. Tortell, B. Rost, and S. Trimborn. 2013. Iron Limitation Modulates Ocean Acidification Effects on Southern Ocean Phytoplankton Communities. *PLOS ONE* **8**: e79890.
- Hu, S. K. and others 2016. Protistan diversity and activity inferred from RNA and DNA at a coastal ocean site in the eastern North Pacific. *FEMS Microbiology Ecology* **92**.
- Kong, H. and others 2023. RNA outperforms DNA-based metabarcoding in assessing the diversity and response of microeukaryotes to environmental variables in the Arctic Ocean. *Science of The Total Environment* **876**: 162608.
- Lampe, R. H. and others 2018. Different iron storage strategies among bloom-forming diatoms. *Proc Natl Acad Sci USA* **115**: E12275-E12284.
- Maldonado, M. T., and N. M. Price. 1999. Utilization of iron bound to strong organic ligands by plankton communities in the subarctic Pacific Ocean. *Deep Sea Research Part II: Topical Studies in Oceanography* **46**: 2447-2473.
- . 2001. Reduction and transport of organically bound iron by *Thalassiosira Oceanica* (Bacillariophyceae). *Journal of Phycology* **37**: 298-310.
- Marchetti, A., and M. T. Maldonado. 2016. Iron, p. 233-279. *In* M. A. Borowitzka, J. Beardall and J. A. Raven [eds.], *The Physiology of Microalgae*. Springer International Publishing.
- Trimborn, S., C. J. M. Hoppe, B. B. Taylor, A. Bracher, and C. Hassler. 2015. Physiological characteristics of open ocean and coastal phytoplankton communities of Western Antarctic Peninsula and Drake Passage waters. *Deep Sea Research Part I: Oceanographic Research Papers* **98**: 115-124.
- Větrovský, T., and P. Baldrian. 2013. The Variability of the 16S rRNA Gene in Bacterial Genomes and Its Consequences for Bacterial Community Analyses. *PLOS ONE* **8**: e57923.
- Wang, X. J., M. Behrenfeld, R. Le Borgne, R. Murtugudde, and E. Boss. 2009. Regulation of phytoplankton carbon to chlorophyll ratio by light, nutrients and temperature in the Equatorial Pacific Ocean: a basin-scale model. *Biogeosciences* **6**: 391-404.